# Synaptotagmin 7 docks synaptic vesicles to support facilitation and Doc2α-triggered asynchronous release

**Zhenyong Wu[1,2†‡], Grant F Kusick[3,4†§], Manon MM Berns[5], Sumana Raychaudhuri[3], Kie Itoh[3], Alexander M Walter[5,6], Edwin R Chapman[1,2*], Shigeki Watanabe[3,7*]**

[1]Department of Neuroscience, University of Wisconsin-Madison, Madison, United States; [2]Howard Hughes Medical Institute, Madison, United States; [3]Department of Cell Biology, Johns Hopkins University, School of Medicine, Baltimore, United States; [4]Biochemistry, Cellular and Molecular Biology Graduate Program, Johns Hopkins University School of Medicine, Baltimore, United States; [5]Department of Neuroscience, University of Copenhagen, Copenhagen, Denmark; [6]Molecular and Theoretical Neuroscience, Leibniz-Institut für Molekulare Pharmakologie, FMP im CharitéCrossOver, Berlin, Germany; [7]Solomon H. Snyder Department of Neuroscience, Johns Hopkins University School of Medicine, Baltimore, United States

**\*For correspondence:**
chapman@wisc.edu (ERC);
shigeki.watanabe@jhmi.edu (SW)

[†]These authors contributed equally to this work

**Present address:** [‡]Shanghai Institute of Materia Medica, Chinese Academy of Sciences, Shanghai, China; [§]University of Wisconsin-Madison, Madison, United States

**Competing interest:** The authors declare that no competing interests exist.

**Abstract** Despite decades of intense study, the molecular basis of asynchronous neurotransmitter release remains enigmatic. Synaptotagmin (syt) 7 and Doc2 have both been proposed as $Ca^{2+}$ sensors that trigger this mode of exocytosis, but conflicting findings have led to controversy. Here, we demonstrate that at excitatory mouse hippocampal synapses, Doc2α is the major $Ca^{2+}$ sensor for asynchronous release, while syt7 supports this process through activity-dependent docking of synaptic vesicles. In synapses lacking Doc2α, asynchronous release after single action potentials is strongly reduced, while deleting syt7 has no effect. However, in the absence of syt7, docked vesicles cannot be replenished on millisecond timescales. Consequently, both synchronous and asynchronous release depress from the second pulse onward during repetitive activity. By contrast, synapses lacking Doc2α have normal activity-dependent docking, but continue to exhibit decreased asynchronous release after multiple stimuli. Moreover, disruption of both $Ca^{2+}$ sensors is non-additive. These findings result in a new model whereby syt7 drives activity-dependent docking, thus providing synaptic vesicles for synchronous (syt1) and asynchronous (Doc2 and other unidentified sensors) release during ongoing transmission.

## eLife assessment

In this **important** study, the authors identify distinct roles for the calcium sensors synaptotagmin 7 and Doc2α in the regulation of asynchronous release and calcium-dependent synaptic vesicle docking in hippocampal neurons. The current work adds to the field by placing the role of the two proteins in a new context, where synaptotagmin 7 acts to promote synaptic vesicle docking and capture after a stimulus, while Doc2α has a role in specifically driving the asynchronous component of release as a calcium sensor. The methods, data, and analyses provide **convincing** support for the conclusions.

## Introduction

Ca²⁺-triggered neurotransmitter release involves at least two kinetically distinct components: one is synchronous, and sets in after tens of microseconds (*Sabatini and Regehr, 1996*) of a stimulus and completes within several milliseconds. The other component is asynchronous; it sets in more slowly and can persist for tens or hundreds of milliseconds (*Goda and Stevens, 1994*; *Kaeser and Regehr, 2014*). The molecular basis of synchronous release has been characterized in depth, and there is broad agreement that the underlying Ca²⁺ sensors are two members of the synaptotagmin (syt) family of proteins, isoforms 1 and 2 (*Stevens and Sullivan, 2003*). The Ca²⁺ sensors that regulate asynchronous release (AR), however, remain a subject of controversy.

AR is thought to play a critical role in information processing and circuit function (*Iremonger and Bains, 2007*; *Kaeser and Regehr, 2014*; *Manseau et al., 2010*) and has been implicated in epileptiform activity in rats and humans (*Jiang et al., 2012*). However, it is impossible to directly test how slow transmission impacts neural networks without knowing what proteins are specifically required for AR.

Two classes of proteins have been proposed as Ca²⁺ sensors for AR in vertebrates: Doc2 and syt7. Two isoforms of Doc2, α and β, function as Ca²⁺ sensors; α mediates AR in hippocampal neurons, and loss of AR in α knockout (KO) neurons can be rescued by β (*Yao et al., 2011*). Both Doc2 and syt7 exhibit the slow Ca²⁺-regulated membrane-binding kinetics and high affinity required for AR (*Hui et al., 2005*; *Xue et al., 2015*; *Yao et al., 2011*). In cultured hippocampal neurons, we previously found that Doc2α drives significant levels of AR both after single action potentials and during trains (*Gaffaney et al., 2014*; *Tu et al., 2023*; *Xue et al., 2018*; *Xue et al., 2015*; *Yao et al., 2011*). This activity depends on Ca²⁺ binding (*Xue et al., 2018*; *Xue et al., 2015*). However, no significant changes in AR were observed in Purkinje cell to deep cerebellar nuclei synapses (*Khan and Regehr, 2020*) and in autaptic cultured hippocampal neurons lacking Doc2 (*Groffen et al., 2010*). These varying results have left the importance of Doc2 for AR in doubt.

Syt7 has received more attention (*Huson and Regehr, 2020*), but its exact role in AR is still uncertain. Syt7 is important for AR during sustained activity at the zebrafish neuromuscular junction (*Wen et al., 2010*). However, conflicting evidence has accumulated for mammalian central synapses. In syt7 KO mouse cortical neurons, no synaptic phenotypes were initially observed (*Maximov et al., 2008*). Later studies found defects in AR when deleting syt7 or preventing it from binding Ca²⁺ in cultured hippocampal neurons (*Bacaj et al., 2013*), in the cerebellum at inhibitory synapses formed between basket cells and Purkinje neurons as well as at granule cell synapses (*Chen et al., 2017*; *Turecek and Regehr, 2018*), and at excitatory synapses between pyramidal cells and Martinotti neurons in the neocortex (*Deng et al., 2020*). However, unlike Doc2α, the role of syt7 in AR is usually only apparent when more than one stimulus is applied. Among these studies, deleting syt7 decreased AR – after a single action potential – only in granule cell synapses (*Turecek and Regehr, 2018*) and in syt1 KO cultured hippocampal neurons (*Bacaj et al., 2013*). In inhibitory synapses of the inferior olive, where release after single action potentials is often highly asynchronous, deleting syt7 actually makes release even slower (*Turecek and Regehr, 2019*). Thus, the importance of syt7 and Doc2α in AR may depend, at least in part, on the frequency of stimulation, the number of stimuli applied, and the synapse under study. The reason for these differences is unknown, reflecting a lack of mechanistic insight into how these proteins support AR.

To directly compare the roles of Doc2α and syt7 in AR, we performed glutamate sensor imaging and electrophysiology experiments in single- and double-knockout (DKO) mouse hippocampal neurons. We then explored how these sensors act on synaptic vesicles using zap-and-freeze electron microscopy and mathematical modeling. In both cultured neurons and acute slices, AR after single action potentials was reduced in Doc2α KOs, but not in syt7 KOs, suggesting that Doc2α more directly contributes to AR. In contrast, AR during train stimulation was reduced to a similar extent in both syt7 and Doc2α KO neurons. Ultrastructural analysis revealed that, in syt7 KO neurons, docked synaptic vesicles fail to recover on millisecond timescales (*Kusick et al., 2020*). Consequently, both synchronous and asynchronous release quickly and profoundly depress during train stimulation. By contrast, in Doc2α KO neurons this syt7-dependent ultrafast recruitment of vesicles was unaffected. Thus, with 50–60% less AR, more newly recruited vesicles are available for synchronous release, leading to slower depression in Doc2α KO neurons. AR in Doc2α /syt7 DKO neurons was similar to Doc2α KO, indicating that the remaining AR is triggered by one or more as-yet-unidentified proteins. Our findings are consistent with a sequential model wherein synaptic vesicle docking, but not fusion, is facilitated by syt7 during

activity. Syt1, Doc2α, and additional sensor(s) act downstream and compete with each other to trigger exocytosis. Overall, we propose that during repetitive stimulation syt7 supplies docked vesicles for synchronous release and Doc2α-triggered AR.

## Results

### Doc2α, but not syt7, triggers asynchronous release in response to single action potentials

To directly monitor glutamate release, we transduced an optical glutamate reporter (iGluSnFR) into cultured mouse hippocampal neurons (*Marvin et al., 2018*; *Marvin et al., 2013*; *Figure 1A and B*, *Figure 1—figure supplement 1*). This approach bypasses postsynaptic factors that influence the time course of transmission as monitored using traditional patch-clamp recordings and allows individual AR events to be measured directly (*Vevea et al., 2021*). We applied single stimuli and monitored the timing of glutamate release at the single-bouton level (*Figure 1C*) from Doc2α knockout (KO, *Doc2a^{tm1Ytk/tm1Ytk}*; *Sakaguchi et al., 1999*) and syt7 KO (*Syt7^{tm1Nan/tm1Nan}*; *Chakrabarti et al., 2003*) neurons and their respective wild-type littermates (WT – *Doc2a^{+/+}* or *Syt7^{+/+}*). We used a medium affinity iGluSnFR (A184V; KD = 40 μM) for these single-stimulus experiments (*Marvin et al., 2018*). Due to the fast activation kinetics of this sensor, compared to its relatively slow relaxation rate (*Marvin et al., 2018*; *Vevea and Chapman, 2020*; *Figure 1—figure supplement 1B*), the timing of each release event was quantified by measuring the time-to-peak of each response. These data were plotted as histograms using 10 ms bins (*Figure 1A and B*, *Figure 1—figure supplement 1*). Peaks detected at the 10 ms time point were defined as the synchronous component of release, and all later peaks were defined as AR events (*Vevea and Chapman, 2020*).

In WT neurons, 64% of release events occurred at the first 10 ms time point (synchronous release: n = 1000 release events); the rest of the events (36%) were defined as asynchronous (*Figure 1C*). This ratio of synchronous to asynchronous release is likely an underestimate since our analysis only counts the number of peaks ('events') and does not take into account multiquantal peaks resulting from near-simultaneous multivesicular release. We have previously determined by quantal analysis that most synchronous peaks after a single action potential are multiquantal, while for AR there are still multiquantal events but they are in the minority (*Vevea et al., 2021*). So in these data, the total amount of synchronous release is underestimated more so than AR; sophisticated quantal analysis using the A184V iGluSnFR recently found the percentage of total release that is AR to be ~25%, with otherwise similar results to ours (*Mendonça et al., 2022*). Nonetheless, this approach faithfully distinguishes synchronous from asynchronous release, as validated by analyzing release from WT neurons treated with EGTA-AM, which reduces AR by chelating residual Ca^{2+} (*Chen and Regehr, 1999*). This treatment greatly reduced AR (*Figure 1C*; 88% synchronous and 12% asynchronous). Syt7 KO neurons had no apparent phenotype (*Figure 1C and D*; 66% synchronous and 34% asynchronous; p=0.39, Dunnett's test). The 2% reduction in AR in the KO was not statistically significant here. However, we note that this small effect is consistent with our earlier measurements using iGluSnFR, which detected a very small but statistically significant decrease in AR in syt7 KO neurons (*Vevea et al., 2021*). By contrast, the asynchronous component was markedly reduced in Doc2α KO neurons (*Figure 1B and C*; 84% synchronous and 16% asynchronous). The peak average fluorescence intensity in both mutants was similar to WT (*Figure 1—figure supplement 1C*), demonstrating that synchronous release is not affected in these mutants (*Vevea et al., 2021*; *Yao et al., 2011*). These findings indicate that Doc2α, but not syt7, plays an important role in AR in response to a single stimulus.

We next turned to acute hippocampal slices to assay Doc2α and syt7 function in a more native circuit. We applied a single stimulus to Schaffer collaterals and measured evoked excitatory postsynaptic currents (EPSCs) from CA1 neurons by whole-cell patch clamp (*Figure 2A*, *Figure 2—figure supplement 1*). As with iGluSnFR imaging in cultured neurons (*Figure 1*), syt7 KOs showed no phenotype – the amplitude and kinetics of EPSCs were nearly identical to those of WT (*Figure 2B–D*). By contrast, in Doc2α KO, the EPSC amplitude was unchanged, but the total charge transfer was reduced by 35% (*Figure 2B–D*). Cumulative charge transfer data were fitted with double exponential functions, reflecting two kinetic components of release (*Figure 2C*; *Goda and Stevens, 1994*). The fast component, representing synchronous release, was similar across all three genotypes. However, the slow, asynchronous component decreased by 61% in Doc2α KO neurons (p<0.0001, Dunnett's test)

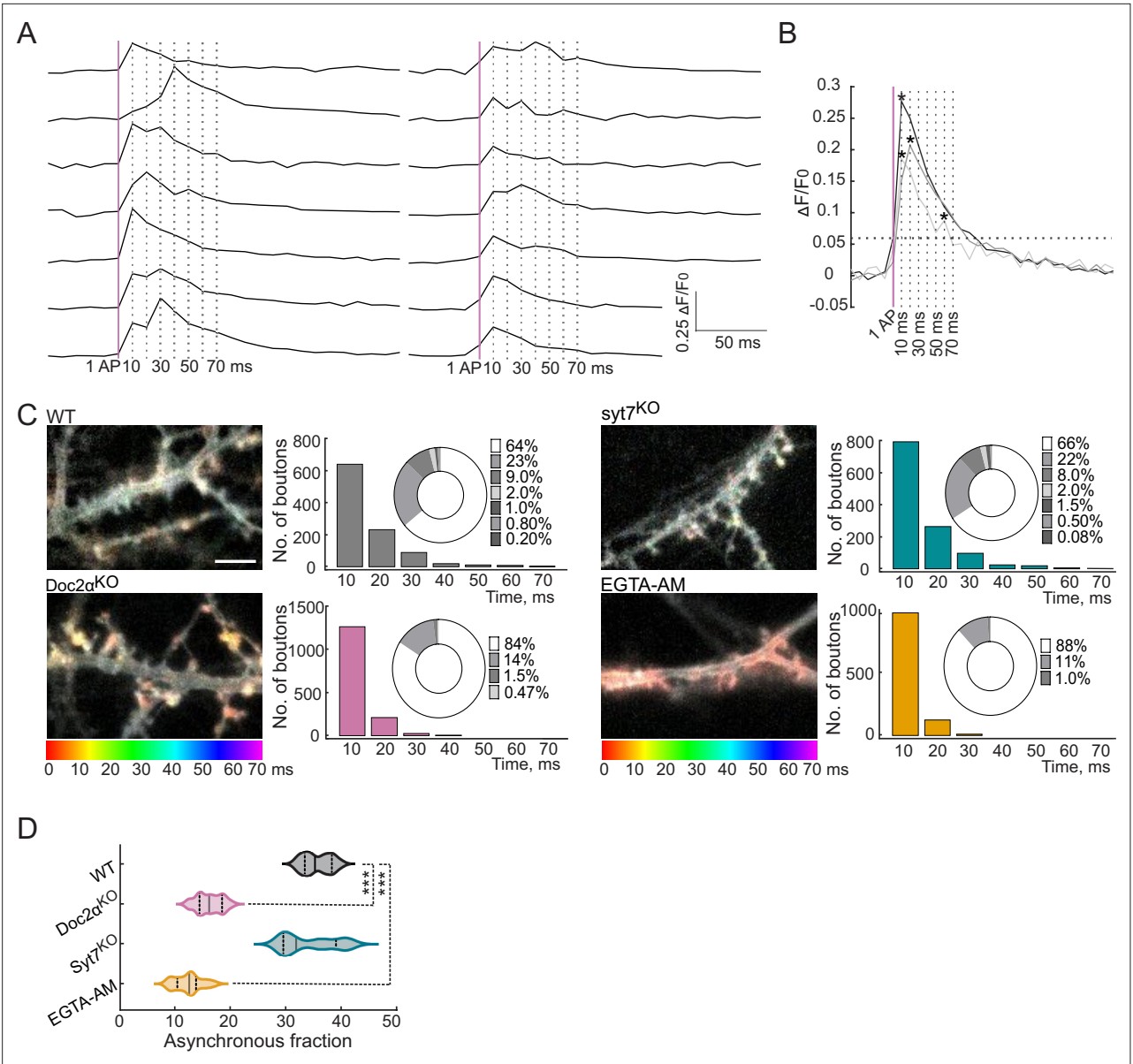

**Figure 1.** Doc2α, and not syt7, drives most of the asynchronous glutamate release triggered by a single action potential (AP) in cultured hippocampal neurons. (**A**) Raw iGluSnFR (A184V) traces from cultured WT mouse hippocampal neurons. A single AP was applied, and images were captured every 10 ms. Peaks that appeared at the first 10 ms time point were defined as synchronous release events; peaks that appeared at later time points were defined as asynchronous release (AR) events. The vertical magenta line indicates the stimulus, and the vertical dashed lines indicate the imaging frames at 10, 20, 30, 40, 50, 60, and 70 ms. (**B**) Three representative traces illustrate synchronous release and AR peaks. The horizontal dashed line is five times the standard deviation of the baseline noise. Individual peaks are indicated by asterisks. Traces of single synchronous (black) and asynchronous peaks (gray), along with a trace in which both kinds of events occurred (light gray), are shown. (**C**) Color coding and binning the temporal changes in iGluSnFR signals within 70 ms after a single stimulus for WT (n = 15 fields of view/coverslips; from four independent litters), Doc2α^KO (n = 17; from five independent litters), syt7^KO (n = 15; from four independent litters), and EGTA-AM-treated neurons (n = 18; from four independent litters). The temporal color code is red (initiation of stimulus) to purple (70 ms after the stimulus); scale bar, 5 μm. Histograms of iGluSnFR (ΔF/F0) peaks were generated using 10 ms binning. Samples were color-coded as follows: WT (gray), Doc2α^KO (reddish), syt7^KO (cyan), and EGTA-AM (orange)-treated neurons. The histograms include donut graph insets to display the fraction of release in each bin. (**D**) Violin plot showing the % AR (release after 10 ms) for each condition. ns are same number of fields of view as in (**C**). Solid bar: median, dotted bars: 25th and 75th percentiles, ends of the violin: minimum and maximum. n = boutons. One-way ANOVA, followed by Dunnett's test, was used to compare mutants against WT. ***p<0.001. Exact p-values: Doc2α^KO: <0.0001, syt7^KO: 0.39, EGTA-AM: <0.0001. All data, summary statistics, test statistics, and p-values are listed in *Figure 1—source data 1*.

The online version of this article includes the following source data and figure supplement(s) for figure 1:

**Figure supplement 1.** Expression of iGluSnFR in cultured hippocampal neurons.

*Figure 1 continued on next page*

*Figure 1 continued*

**Figure supplement 1—source data 1.** Data used to generate *Figure 1—figure supplement 1*.

**Source data 1.** Data used to generate *Figure 1*.

and 10% in syt7 KO neurons (*Figure 2D*; p=0.15), though this latter difference was not statistically significant. EGTA-AM treatment of WT slices reduced the slow component by 74% (*Figure 2B–D*; p<0.0001). Finally, the time constants for both components of evoked EPSCs were similar across all genotypes (*Figure 2D*). Together, these data further indicated that Doc2α, but not syt7, is required for a large fraction of AR after single action potentials in excitatory mouse hippocampal synapses.

## Both Doc2α and syt7 are important for asynchronous release after repeated action potentials

We then addressed the roles of Doc2α and syt7 in AR during train stimulation. Fifty stimuli, at 10 and 20 Hz, were applied to cultured hippocampal neurons that expressed iGluSnFR (*Figure 3A and B*). To temporally resolve the response from each pulse during the train, we used a faster, lower-affinity version of iGluSnFR (S72A; KD = 200 µM; *Marvin et al., 2018*; *Marvin et al., 2013*). These faster kinetics were essential to resolve individual peaks during train stimulation; however, this sensor

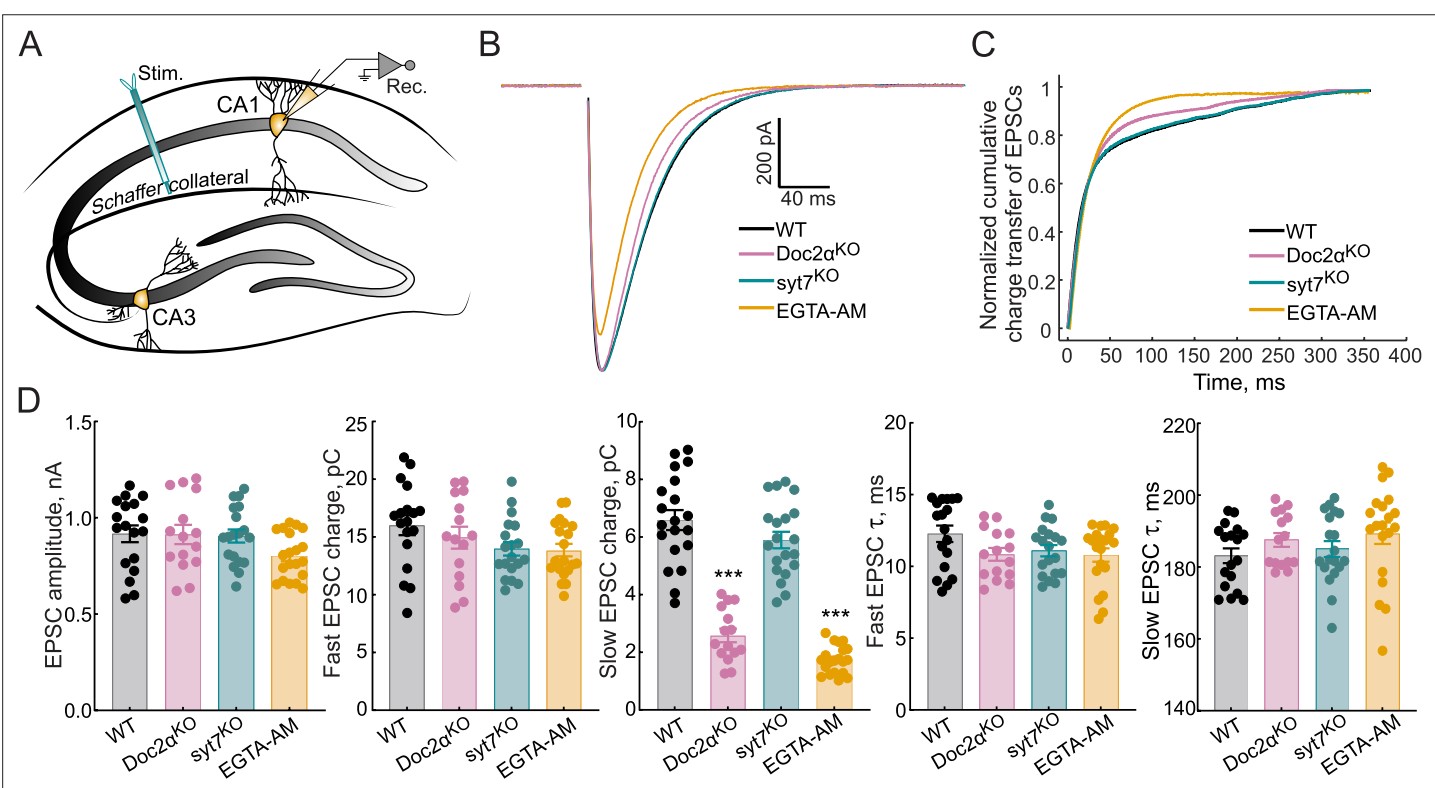

**Figure 2.** Doc2α, and not syt7, drives most of the asynchronous release (AR) triggered by a single action potential at Schaffer collaterals. (**A**) Schematic of the recording scheme of hippocampal slices showing stimulation of the Schaffer collateral fibers and whole-cell patch-clamp recordings of CA1 pyramidal neurons. (**B**) Averaged traces of evoked excitatory postsynaptic currents (EPSCs) recorded from WT (n = 18 recordings; from five independent litters), Doc2α[KO] (n = 15; from four independent litters), syt7[KO] (n = 19; from six independent litters), and EGTA-AM-treated neurons (n = 20; from four independent litters). (**C**) Normalized cumulative EPSC charge transfer over 400 ms. (**D**) Bar graphs, from corresponding groups in panels (**B–C**), summarizing the amplitude, fast and slow EPSC charge, and the fast and slow EPSC *t*-values. Error bars are mean ± standard error of the mean. One-way ANOVA, followed by Dunnett's test, was used to compare mutants against WT. ** and *** indicate p< 0.01 and p<0.001. Exact p-value for syt7 slow EPSC charge = 0.15. All data, summary statistics, and p-values are listed in *Figure 2—source data 1*.

The online version of this article includes the following source data and figure supplement(s) for figure 2:

**Figure supplement 1.** Recording setup for electrophysiology of acute hippocampal slices.

**Source data 1.** Data used to generate *Figure 2*.

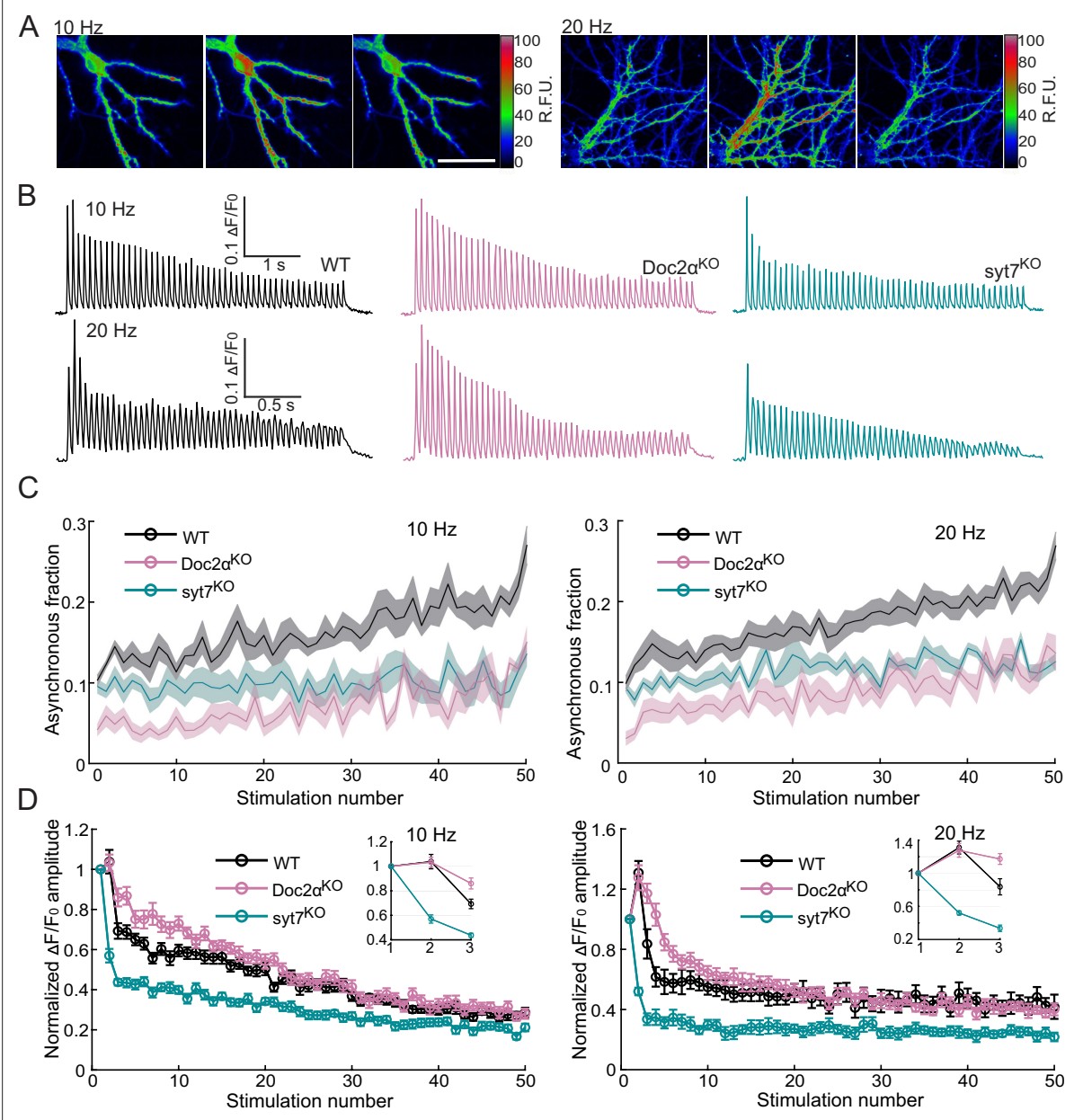

**Figure 3.** Both Doc2α and syt7 contribute to asynchronous glutamate release during train stimulation but have opposite effects on short-term plasticity in cultured hippocampal neurons. (**A**) Representative iGluSnFR images from trains of 50 action potentials (APs) at 10 Hz (left) or 20 Hz (right). Images on the left, middle, and right each show a sum of 20 frames (200 ms) before, during, and after stimulation, respectively; scale bar, 40 μm. (**B**) Representative iGluSnFR responses from WT (10 Hz: n = 12 fields of view/coverslips; from three independent litters; 20 Hz: n = 17; from four independent litters), Doc2α$^{KO}$ (n = 14; from four independent litters; 20 Hz: n = 16; from four independent litters), and syt7$^{KO}$ neurons (n = 15; from three independent litters; 20 Hz: n = 12; from three independent litters), stimulated at 10 Hz (top) and 20 Hz (bottom). (**C**) The asynchronous release (AR) fraction was defined as iGluSnFR peaks that occurred after the first 10 ms time point (i.e., from the 20 ms time point onward), divided by the total number of peaks that occurred at all time points between each stimulus. These data are plotted as a function of the stimulus number. Mean values (bold lines) ± standard error of the mean (shaded regions) are indicated. (**D**) Plot of the normalized amplitude (with 1 being the amplitude of the first response) of iGluSnFR signals evoked by 50 stimuli at 10 and 20 Hz. Dots and error bars are mean ± standard error of the mean. Insets show the first three points of each trace on expanded scales.

The online version of this article includes the following source data and figure supplement(s) for figure 3:

**Source data 1.** Data used to generate *Figure 3*.

**Figure supplement 1.** Additional raw iGluSnFR traces during 20 Hz train stimulation, and paired-pulse ratios from the first two responses at 10 and 20 Hz.

*Figure 3 continued on next page*

*Figure 3 continued*

**Figure supplement 1—source data 1.** Data used to generate *Figure 3—figure supplement 1*.

**Figure supplement 2.** Spatial distribution of asynchronous release (AR), measured via iGluSnFR imaging, in cultured hippocampal neurons.

**Figure supplement 2—source data 1.** Data used to generate *Figure 3—figure supplement 2*.

underestimates the fraction of AR (~10% of total release for a single action potential) compared to the A184V variant used above that overestimates the fraction of AR (~35% of total release for a single action potential). This is because S72A is less sensitive and misses many uniquantal events; as discussed above, our analysis quantifies release by number of peaks, and most synchronous peaks after a single action potential are multiquantal, while most AR peaks are uniquantal (*Vevea et al., 2021*). Still, the S72A variant reported the same phenotypes as the A184V variant after the first action potential (*Figure 3B and C*).

At both 10 and 20 Hz, WT neurons exhibited slight facilitation of synchronous release (measured as the peak average fluorescence intensity) in response to the second pulse, and thereafter depressed as the train went on, eventually reaching a steady state (*Figure 3B and D*, *Figure 3—figure supplement 1*). In addition, the asynchronous fraction gradually increased during the train at both stimulation frequencies (*Figure 3C*, *Figure 3—figure supplement 1*). We examined the first two responses in the train and found that paired-pulse facilitation was similar between WT and Doc2α KO neurons (WT: 1.04 ± 0.06, Doc2α KO: 1.03 ± 0.04, p=0.9953, with an interstimulus interval ($\Delta t$) = 100 ms; WT: 1.31 ± 0.08, Doc2α KO: 1.28 ± 0.08, p>0.9191, $\Delta t$ = 50 ms, Dunnett's test; *Figure 3—figure supplement 1*). However, in KO neurons, synaptic depression set in more slowly from the third action potential onward, eventually reaching the same steady-state depression as WT later in the train. This is consistent with the idea that the sensors for synchronous and asynchronous release compete with each other, so that reducing AR promotes sustained synchronous release (*Otsu et al., 2004*; *Yao et al., 2011*). In contrast, syt7 KO neurons displayed paired-pulse depression instead of facilitation (syt7 KO: 0.57 ± 0.03, p<0.0001, $\Delta t$ = 1001 ms; syt7 KO: 0.52 ± 0.03, p<0.0001, $\Delta t$ = 50 ms, Dunnett's test; *Figure 3—figure supplement 1*), as well as faster and more severe synaptic depression as the train progressed (*Figure 3D*), consistent with previous reports (*Chen et al., 2017*; *Jackman et al., 2016*; *Liu et al., 2014*; *Vevea et al., 2021*). As in the single-stimulus measurements (*Figure 1*), AR was decreased in Doc2α KOs, but not syt7 KOs, in response to the first pulse in the stimulus train. However, from the second action potential onward, both KOs yielded reductions in AR compared to WT, such that by the end of the train the fraction of AR was reduced by 53 and 50% in syt7 and Doc2α KO neurons, respectively (*Figure 3A–C*; see also *Vevea et al., 2021*).

One potential source of the observed phenotype in syt7 KO during the train is variability among synapses. That is, the boutons that respond to the second pulse onward may have different release properties. To test such a possibility, we analyzed the distribution of the iGluSnFR signals from 923 boutons. We did not observe spatial segregation of AR during the first two responses (see *Figure 3—figure supplement 2* for further analysis and details). Thus, the differences in AR between the first and second responses are not due to the differences regarding which boutons responded during the first and second stimuli. Together, these data suggest that both Doc2α and syt7 contribute to AR from the second pulse onward during the train.

Next, we extended the train stimulation experiments to hippocampal slices, via patch-clamp recordings (*Figure 4A*), and observed the same trends that were obtained using iGluSnFR in cultured neurons (*Figure 3*). Namely, during 20 Hz stimulation, peak amplitudes (synchronous release) depressed more slowly in Doc2α KO, and syt7 KO neurons had faster and more profound synaptic depression during the train, compared to WT (*Figure 4B*). Interestingly, loss of syt7 caused more severe synaptic depression than EGTA-AM. To estimate the extent of AR in these electrophysiological experiments, we calculated the tonic charge transfer, as previously described (*Otsu et al., 2004*; *Yao et al., 2011*). This approach revealed that syt7 and Doc2α KO neurons both exhibited 65% reductions in the tonic charge transfer component, while EGTA-AM treatment reduced tonic charge by 74% (*Figure 4C*). These data are all consistent with both $Ca^{2+}$-binding proteins, Doc2α and syt7, playing important roles in AR driven by residual $Ca^{2+}$ (*Wen et al., 2010*; *Yao et al., 2011*).

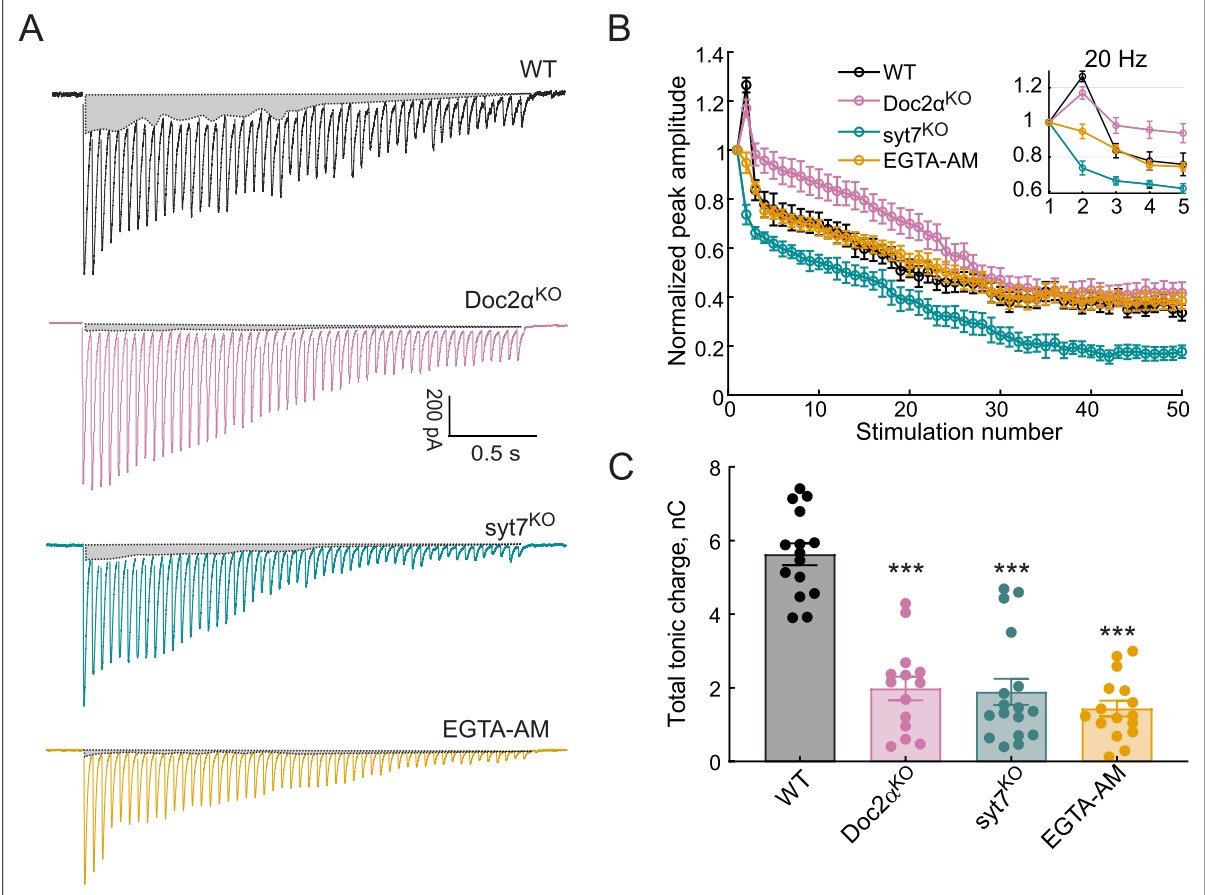

**Figure 4.** Both Doc2α and syt7 contribute to asynchronous release (AR) during train stimulation but have opposite effects on short-term plasticity at Schaffer collaterals. (**A**) Representative excitatory postsynaptic currents (EPSCs) triggered by 20 Hz stimulus trains using WT, Doc2α$^{KO}$, syt7$^{KO}$, and EGTA-AM-treated neurons. The shaded gray areas indicate the tonic charge component that likely corresponds to AR. (**B**) The peak amplitude of each EPSC during the train was normalized to the first EPSC from WT (n = 12 recordings; from four independent litters), Doc2α$^{KO}$ (n = 11; from three independent litters), syt7$^{KO}$ (n = 10; from three independent litters), and EGTA-AM-treated neurons (n = 11, from three independent litters). Dots and error bars are mean ± standard error of the mean. (**C**) Bar graph showing the total tonic charge transfer (indicated by the gray areas in (**A**)), as a measure of cumulative AR from the train, from WT, Doc2α$^{KO}$, syt7$^{KO}$, and EGTA-AM-treated hippocampal slices. Error bars are mean ± standard error of the mean. One-way ANOVA, followed by Dunnett's test, was used to compare mutants against WT. \*\*\*p<0.001. All data, summary statistics, and p-values are listed in *Figure 4—source data 1*.

The online version of this article includes the following source data for figure 4:

**Source data 1.** Data used to generate *Figure 4*.

## Doc2α mediates asynchronous fusion, while syt7 docks vesicles

Why is Doc2α important for AR after single and multiple stimuli, while syt7 only contributes starting with a second action potential? And why is short-term plasticity so strongly disrupted in syt7 KOs? Recent studies have proposed that synaptic vesicles rapidly transition from undocked to docked states during activity, and that this could, in principle, provide a common basis for AR, synaptic facilitation, and resistance to synaptic depression (*Chang et al., 2018*; *Kobbersmed et al., 2020*; *Kusick et al., 2022*; *Miki et al., 2018*; *Miki et al., 2016*; *Neher and Brose, 2018*; *Silva et al., 2021*). In this model, after Ca$^{2+}$ influx, initially undocked and release-incompetent vesicles can dock and fuse ('two-step' release: *Blanchard et al., 2020*; *Miki et al., 2018*), resulting in AR, or dock and then fuse after another action potential, leading to facilitation and resistance to depression (*Miki et al., 2016*; *Neher and Brose, 2018*). Consistent with this notion, we previously found that a substantial fraction of vesicles undock in response to an action potential, and enter into a dynamic state in which they re-dock, within 15 ms, to replenish vacated release sites. We termed this process 'transient docking' since vesicles again undock over the next 100 ms (*Kusick et al., 2020*). Importantly, like AR and facilitation,

transient docking was blocked by EGTA-AM. Thus, our imaging and electrophysiology findings could be explained by these two $Ca^{2+}$ sensors playing distinct roles in docking and fusion, with Doc2α triggering AR, while syt7 mediates transient docking.

To test this hypothesis, we performed zap-and-freeze electron microscopy experiments (*Kusick et al., 2020*). This approach allows us to measure synchronous and asynchronous vesicle fusion, as well as depletion and replenishment of docked vesicles, in the same experiment. Vesicles that appear to be in contact with the plasma membrane are considered docked. Unstimulated syt7 and Doc2α KO synapses appeared qualitatively normal (see *Figures 5A–D* and *6A–D*, *Figure 5—figure supplement 1* for example micrographs). Knocking out either protein did not affect the number of docked vesicles at rest (*Figure 6D and E*) nor the number and distribution of synaptic vesicles within 100 nm of the plasma membrane at the active zone (*Figure 5—figure supplement 2D*). Therefore, the physiological defects in syt7 and Doc2α KO synapses cannot be accounted for by any ultrastructural phenotypes at rest.

We then assessed synaptic vesicle exocytosis by quantifying the appearance of exocytic pits following a single action potential (1 ms, 1.2 mM external $Ca^{2+}$, 37°C) and freezing the neurons at 5, 11, and 14 ms after the action potential. These time points capture synchronous fusion, asynchronous fusion, and transient docking, respectively (*Kusick et al., 2020*; *Li et al., 2021*). Consistent with our previous findings (*Kusick et al., 2020*; *Li et al., 2021*), in WT controls exocytic pits peaked at 5 ms after an action potential, declined to less than half of this peak at 11 ms, then further declined to baseline by 14 ms (; pits in the active zone per synaptic profile: syt7 WT: no stim: 0.03 pits; 5 ms; 0.22 pits; 11 ms: 0.13 pits; 14 ms: 0.04 pits; Doc2α WT: no stim: 0.021 pits; 5 ms: 0.35 pits; 11 ms: 0.08 pits; 14 ms: 0.04 pits; see *Figure 5—figure supplement 2A* for active zone sizes from each condition and *Figure 5—figure supplement 2B* for pit data normalized to the size of active zones, and *Figure 5—source data 2* for full descriptive statistics and test statistics). In syt7 KOs, this sequence was unchanged: the number of pits at each time point in the KO was not different from the corresponding WT control (pits per synaptic profile: syt7 KO: no stim: 0.04 pits; 5 ms: 0.28 pits; 11 ms: 0.14 pits; 14 ms: 0.06 pits; p>0.9 for each compared to the corresponding syt7 WT time point, Brown–Forsythe ANOVA with Games–Howell post hoc test). By contrast, in Doc2α KO synapses, the number of pits at 5 ms was unchanged, but pits at 11 ms were almost completely abolished (pits per synaptic profile: Doc2α KO: no stim: 0.02 pits; 5 ms: 0.32 pits; 11 ms: 0.03 pits; 14 ms: 0.01 pits; p=0.04 between WT and KO 11 ms, p=0.02 between WT no stim and 11 ms, p>0.9 between KO no stim and 11 ms, Brown–Forsythe ANOVA with Games–Howell post hoc test). This phenotype mirrors the effect of EGTA-AM we previously reported (*Kusick et al., 2020*), consistent with Doc2α driving a large fraction of residual $Ca^{2+}$-dependent AR. In sum, we obtained the same result from three different techniques (iGluSnFR imaging, patch-clamp electrophysiology, and zap-and-freeze electron microscopy): after a single action potential, Doc2α accounts for 54–67% of AR at hippocampal excitatory synapses, whereas deleting syt7 has no effect.

Is syt7 required for activity-dependent transient docking of synaptic vesicles? To test this possibility, we quantified all vesicles within 100 nm of the plasma membrane at the active zone. The distribution of undocked vesicles did not detectably change in any of the genotypes, nor at any of the time points after stimulation (*Figure 5—figure supplement 2D*). WT controls showed rapid depletion and replenishment of docked vesicles after an action potential, as previously reported (*Kusick et al., 2020*; *Figure 6E and F*). At 5 and 11 ms, there were ~40% fewer docked vesicles than in the no stimulation controls, then at 14 ms docked vesicles returned to baseline (*Figure 6E and F*; docked vesicles per synaptic profile: syt7 WT: no stim: 2.19; 5 ms; 1.52; 11 ms: 1.36; 14 ms: 1.93; Doc2α WT: no stim: 1.90; 5 ms: 0.91; 11 ms: 1.00; 14 ms: 1.62; see *Figure 5—figure supplement 2A* for active zone sizes from each condition and *Figure 5—figure supplement 2C* for docked vesicle data normalized to the size of active zones, and *Figure 5—source data 2* for full descriptive statistics and test statistics). In syt7 KO neurons, consistent with our previous data (*Vevea et al., 2021*), there were ~25% fewer docked vesicles than in WT controls at 5 ms and 11 ms. At 14 ms, docked vesicles completely failed to recover in syt7 KO, remaining at less than half of the no-stim control (docked vesicles per synaptic profile: syt7 KO: no stim: 2.12; 5 ms: 1.07; 11 ms: 1.07; 14 ms: 1.09; p<0.0001 between WT and KO at 14 ms, p=0.001 at 5 ms between WT and KO, p=0.16 at 5 ms between WT and KO, Brown–Forsythe ANOVA with Games–Howell post hoc test). These data indicate that syt7 is absolutely required for transient docking of synaptic vesicles.

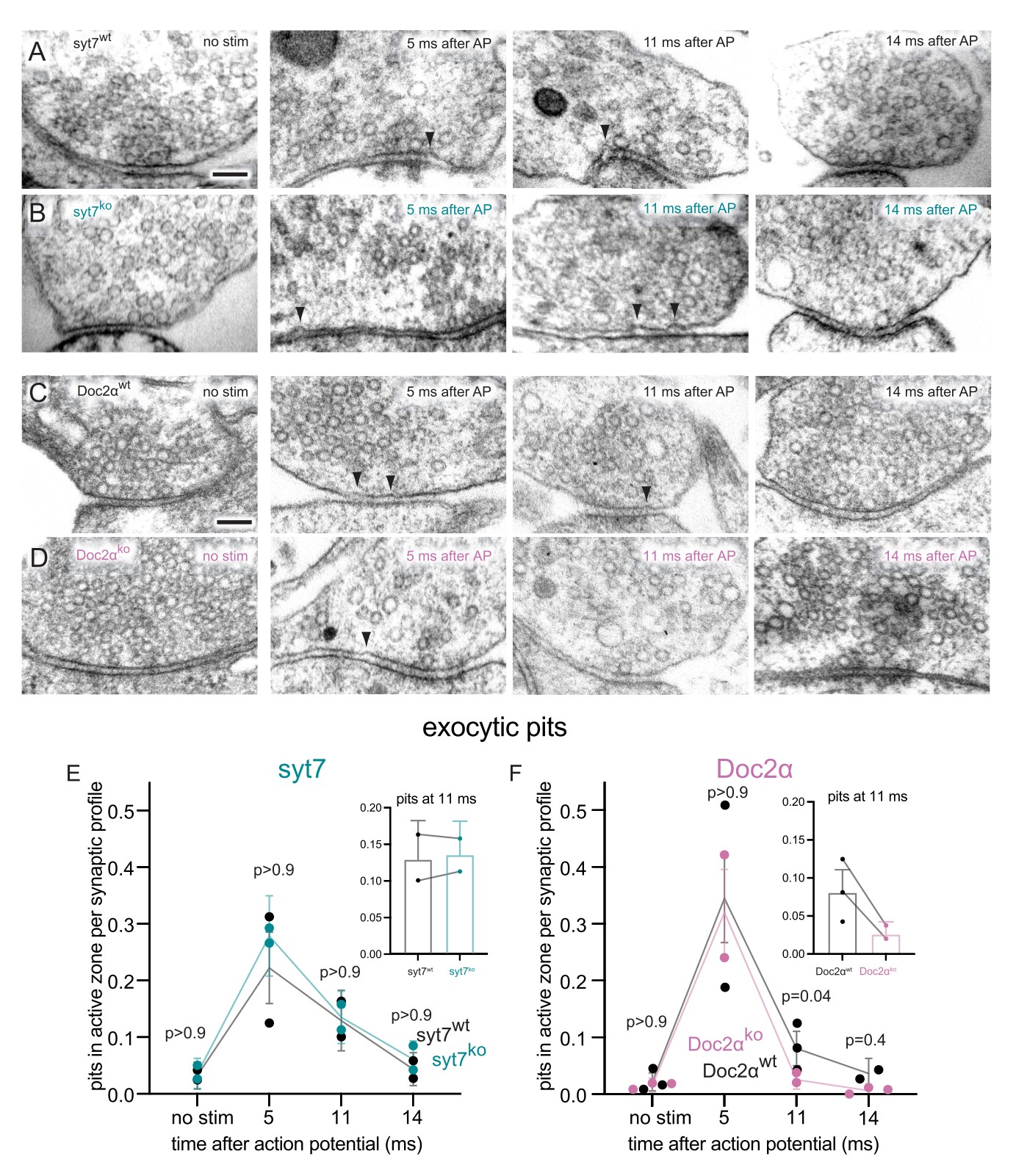

**Figure 5.** Doc2α, but not syt7, is important for asynchronous synaptic vesicle fusion triggered by a single action potential (AP) in cultured hippocampal neurons. (**A–D**) Example transmission electron micrographs of syt7^WT (**A**), syt7^KO (**B**), Doc2α^WT (**C**), and Doc2α^KO (**D**) synapses frozen at the indicated time points after an AP or without stimulation (no stim). Arrows indicate pits in the active zone, which are presumed to be synaptic vesicles that have fused with the plasma membrane. (**E**) The number of pits in the active zone per synaptic profile (part of the synapse captured in a 2D section) in syt7 wild-type

*Figure 5 continued*

littermate controls and knockouts. Error bars indicate 95% confidence interval of the mean for all data pooled together; each dot indicates the mean from a single biological replicate. Inset: same data from the 11 ms time point from each genotype, with lesser y-axis range; lines indicate data are from the same biological replicate (experiment), *not* that they are paired. p-Values are from comparisons between wild-type controls (nostim, n = 217; 5 ms, n = 216; 11 ms, n = 217; 14 ms, n = 230 synaptic profiles) and knockouts (no stim, n = 193; 5 ms, n = 215; 11 ms, n = 230; 14 ms, n = 212 synaptic profiles) frozen at the same time point. (F) Same as (E), but for Doc2α wild-type controls (no stim, n = 327; 5 ms, n = 229; 11 ms, n = 336; 14 ms, n = 192 synaptic profiles) and knockouts (no stim, n = 330; 5 ms, n = 231; 11 ms, n = 352; 14 ms, n = 321 synaptic profiles). Number of biological replicates (from separate cultures frozen on different days and analyzed in separate randomized batches): 2 for each of the syt7 time points, 2 for 5 ms and 14 ms Doc2 WT, 2 for 5 ms and 11 ms Doc2 KO, 3 for Doc2 WT no stim and 11 ms, 3 for Doc2 KO no stim and 14 ms. The number of pits between wild type and knockout was compared using Brown–Forsythe ANOVAs with post hoc Games–Howell's multiple-comparisons tests. In pairwise tests, the same time point for wild type vs. knockout as well as each time point within each genotype were compared against each other. Only comparisons between the same time point for wild type vs. knockout are shown. See *Figure 5—source data 2* for all pairwise comparisons, summary statistics, and test statistics. See *Figure 5—figure supplement 2A* for active zone sizes from each condition and *Figure 5—figure supplement 2B* for pit data normalized to the size of active zones. See *Figure 5—figure supplement 1* for more example micrographs. See *Figure 5—source data 1* for all data used to generate the figures.

The online version of this article includes the following source data and figure supplement(s) for figure 5:

**Source data 1.** Data used to generate the graphs in *Figure 5*.

**Source data 2.** Summary statistics and full output of statistical tests performed on the data in *Figure 5*.

**Figure supplement 1.** Additional example transmission electron micrographs.

**Figure supplement 2—source data 1.** Data used to generate the graphs in *Figure 5—figure supplement 2*.

**Figure supplement 2.** Additional quantification from electron microscopy.

---

In Doc2α KO neurons, by contrast, we observed normal docked vesicle depletion and faster recovery (*Figure 6F*). The number of docked vesicles was 40% greater in the KO than in the WT at 11 ms, about halfway toward full recovery to baseline (docked vesicles per synaptic profile: Doc2α KO: no stim: 1.85; 5 ms; 1.11; 11 ms: 1.40; 14 ms: 1.63; p<0.0001 between WT and KO at 11 ms, p<0.0001 between 11 ms and no stim in the KO, Brown–Forsythe ANOVA with Games–Howell post hoc test). This is consistent with the idea that attenuation of AR spares some vesicles for ongoing synchronous release (*Otsu et al., 2004*; *Yao et al., 2011*), which could in part explain the slower train depression in Doc2α KOs (*Figures 3 and 4*). We conclude that transient docking completely fails in the absence of syt7, while Doc2α is dispensable for docked vesicle recovery. Taken together with our data on exocytosis, these results suggest that syt7 supports synchronous and asynchronous release after multiple stimuli by triggering transient docking, while Doc2α is a Ca²⁺ sensor for asynchronous fusion itself.

## Model for sequential action of syt7 and Doc2α during asynchronous release

To test whether a reaction scheme in which syt7 and Doc2α act sequentially is consistent with the physiological (*Figures 1–4*) and morphological (*Figures 5 and 6*) phenotypes observed in this work, we constructed a mathematical model. We aimed to investigate whether the theory proposed above could replicate the experimental data, though we cannot rule out that other unexplored model implementations match the experimental data similarly well. In the model, syt7 promotes Ca²⁺-dependent synaptic vesicle docking and Doc2α is a high-affinity Ca²⁺-sensor for synaptic vesicle fusion (see *Figure 7A* for simplified scheme, *Figure 7—figure supplement 1* for full reaction scheme, and *Figure 7—source code 1* for all underlying code). As in previously proposed models (*Miki et al., 2016*; *Neher and Brose, 2018*; *Pan and Zucker, 2009*; *Walter et al., 2013*; *Weingarten et al., 2022*), we assumed docking of synaptic vesicles to occur in two steps: first, synaptic vesicles are tethered to release sites before they dock and become fusion competent (*Figure 7A*). Vesicle exocytosis in the model depends on the total number of Ca²⁺ ions bound to syt1 and Doc2α per vesicle/release site, and is simulated using an adaptation of previously developed dual-sensor models (*Kobbersmed et al., 2020*; *Sun et al., 2007*). In this model, each Ca²⁺ ion bound to either sensor was assumed to lower the energy barrier for vesicle fusion. This means that the rate of fusion increases exponentially as more Ca²⁺ ions are bound to syt1 and Doc2α (*Schotten et al., 2015*). We assumed that the fusion rate is accelerated more when Ca²⁺ binds syt1 than Doc2α. In our model, we implemented the action of syt1 on the vesicular fusion rate following the five-site binding model (*Lou et al., 2005*), which captures the physiological Ca²⁺ dependence of fast neurotransmitter release without explicit modeling of molecular interactions (*Kobbersmed et al., 2022*). The additional role of Doc2α was

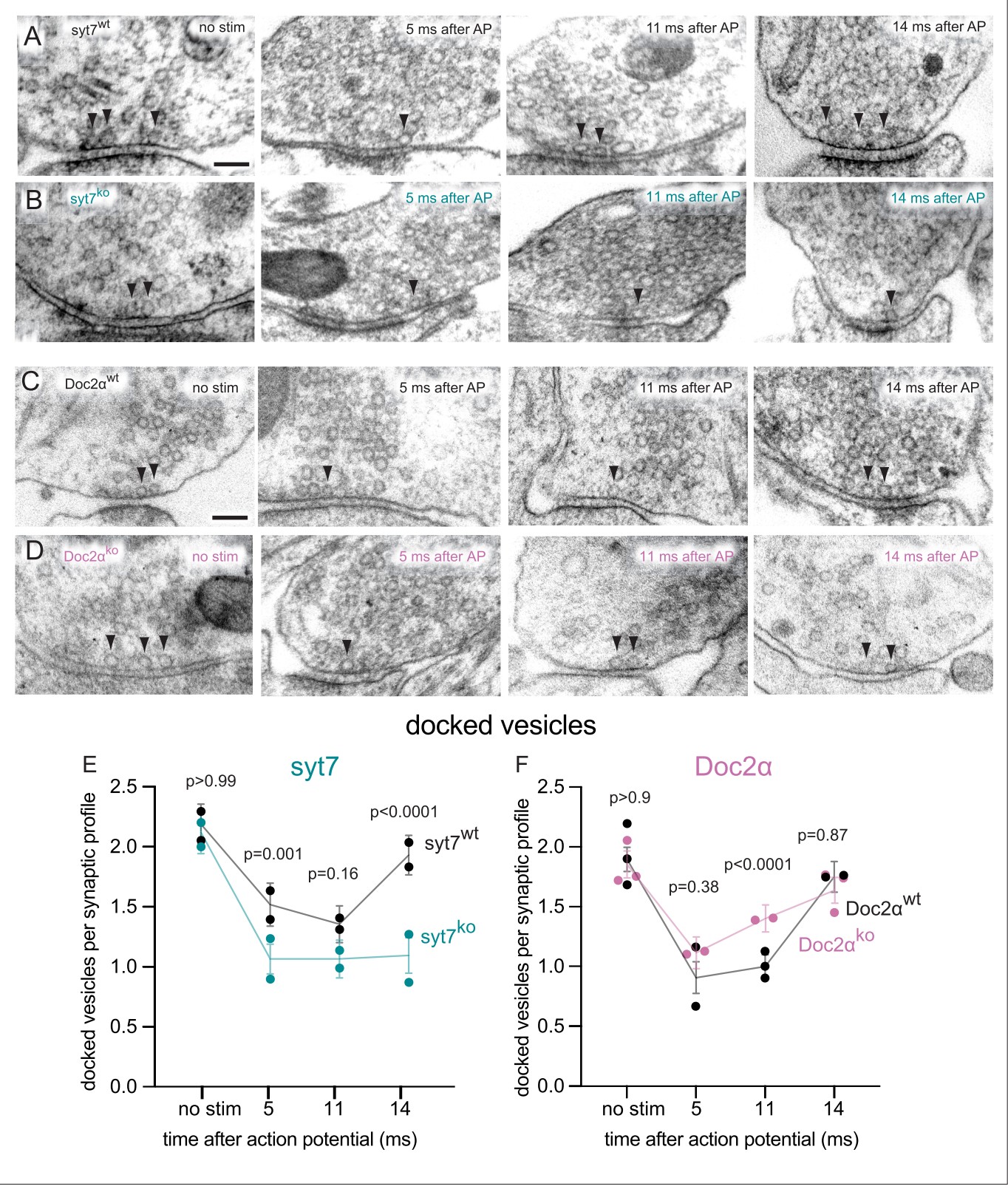

**Figure 6.** Syt7 is absolutely required for transient docking, while docked vesicles recover more quickly without Doc2α. (**A–D**) Example transmission electron micrographs of syt7[WT] (**A**), syt7[KO] (**B**), Doc2α[WT] (**C**), and Doc2α[KO] (**D**) synapses frozen at the indicated time points after an action potential (AP) or without stimulation (no stim). Arrows indicate docked vesicles (synaptic vesicles with no visible distance between their membrane and the plasma membrane). (**E**) The number of docked vesicles per synaptic profile (part of the synapse captured in a 2D section) in syt7 wild-type littermate controls

*Figure 6 continued on next page*

*Figure 6 continued*

and knockouts. Error bars indicate 95% confidence interval of the mean for all data pooled together; each dot indicates the mean from a single biological replicate. p-Values are from comparisons between wild-type controls (no stim, n = 217; 5 ms, n = 216; 11 ms, n = 217; 14 ms, n = 230 synaptic profiles) and knockouts (no stim, n = 193; 5 ms, n = 215; 11 ms, n = 230; 14 ms, n = 212 synaptic profiles) frozen at the same time point. (F) same as (E), but for Doc2α wild-type controls (no stim, n = 327; 5 ms, n = 229; 11 ms, n = 336; 14 ms, n = 192 synaptic profiles) and knockouts (no stim, n = 330; 5 ms, n = 231; 11 ms, n = 352; 14 ms, n = 321 synaptic profiles). Number of biological replicates (from separate cultures frozen on different days and analyzed in separate randomized batches): 2 for each of the syt7 time points, 2 for 5 ms and 14 ms Doc2 WT, 2 for 5 ms and 11ms Doc2 KO, 3 for Doc2 WT no stim and 11 ms, and 3 for Doc2 KO no stim and 14 ms. The numbers of pits between wild type and knockout were compared using Brown–Forsythe ANOVAs with post hoc Games–Howell's multiple-comparisons tests. In pairwise tests, the same time point for wild type vs. knockout as well as each time point within each genotype were compared against each other. Only comparisons between the same time point for wild type vs. knockout are shown. See *Figure 6—source data 2* for all pairwise comparisons, summary statistics, and test statistics. See *Figure 5—figure supplement 2A* for active zone sizes from each condition, *Figure 5—figure supplement 2C* for docked vesicle data normalized to the size of active zones, and *Figure 5—figure supplement 2D* for distribution of undocked vesicles. See *Figure 5—figure supplement 1* for more example micrographs. See *Figure 6—source data 1* for all data used to generate the figures.

The online version of this article includes the following source data for figure 6:

**Source data 1.** Data used to generate the graphs in *Figure 6*.

**Source data 2.** Summary statistics and full statistical test output for the data in *Figure 6*.

implemented by two additional, independent Ca²⁺-binding sites, following previously developed dual-sensor models (**Kobbersmed et al., 2020**; **Sun et al., 2007**). This was chosen because deleting the fast syt in the Calyx of Held uncovered a slow Ca²⁺-dependent fusion reaction dependence with a cooperativity of two (**Sun et al., 2007**). We assumed that both proteins induce synaptic vesicle exocytosis by acting through a common substrate (e.g., a limited number of SNARE complexes or plasma membrane-binding sites; **Groffen et al., 2010**), which was not explicitly included in the model. The competition between Doc2α and syt1 was implemented by limiting the total number of Ca²⁺ ions that could be bound to syt1 and Doc2α to five ions per release site/synaptic vesicle.

Like Doc2α and syt1, which reduce the energy barrier for exocytosis upon Ca²⁺ binding, we assumed that syt7 acts as a catalyst for docking (**Walter et al., 2013**). Because docking, unlike fusion, is reversible (**Kusick et al., 2020**; **Murthy and Stevens, 1999**; **Yavuz et al., 2018**), a reduction in the barrier height alone (without a change in the free energies of the docked or undocked states) increases both the docking and the undocking rates. Such a catalytic function of syt7 is consistent with the observations that (1) its loss did not affect steady-state docking (**Figure 6**) and that (2) during transient docking in cultured hippocampal neurons, the number of docked vesicles returns to the pre-stimulation baseline. However, it is possible that Ca²⁺-bound syt7 instead selectively increases the docking rate or decreases the undocking rate (and such a mechanism may explain its role in facilitation: **Kobbersmed et al., 2020**; **Lin et al., 2022**; **Pan and Zucker, 2009**; **Weingarten et al., 2022**).

Since both Doc2α and syt7 are present on the plasma membrane (**Sugita et al., 2001**; **Verhage et al., 1997**; **Vevea et al., 2021**; **Weber et al., 2014**), they were assumed to bind Ca²⁺ irrespective of whether the site is occupied by a vesicle (see 'Methods'). In this way, the release site (syt7 and Doc2α) maintains its Ca²⁺-bound state independent of whether synaptic vesicles tether, dock, or fuse. This is highly relevant for Ca²⁺ sensors with high affinity but slow binding kinetics, like syt7 and Doc2α (**Hui et al., 2005**; **Xue et al., 2015**; **Yao et al., 2011**) and distinguishes them from the vesicular sensor syt1, which is unlikely to bind Ca²⁺ on undocked synaptic vesicles (**Bai et al., 2004**; **Radhakrishnan et al., 2009**).

In our model, we explicitly calculate Ca²⁺ association with and dissociation from syt1, syt7, and Doc2α. This causes variable reaction rates for synaptic vesicle docking, undocking, and fusion, resulting in a large increase in the number of states a release site can be in and in the number of model parameters. Nevertheless, we decided on explicit simulation of Ca²⁺ binding to these proteins as this allowed us to integrate information on their Ca²⁺-binding properties. Moreover, the explicit simulation of Ca²⁺ binding and unbinding is highly relevant in systems with rapid dynamics as it takes time before the Ca²⁺ sensors reach equilibrium. This especially holds for the slow Ca²⁺ sensors syt7 and Doc2α. The affinity of syt1 and syt7 as well as the Ca²⁺-binding rate of syt1 and its effect on the energy barrier were estimated from the literature (see *Table 1*). For simplicity, Doc2α and syt7 were assumed to share the same Ca²⁺-binding dynamics and that the docking/undocking rate is modulated by the association of Ca²⁺ to two syt7-binding sites at each release site. These simplifications reduced

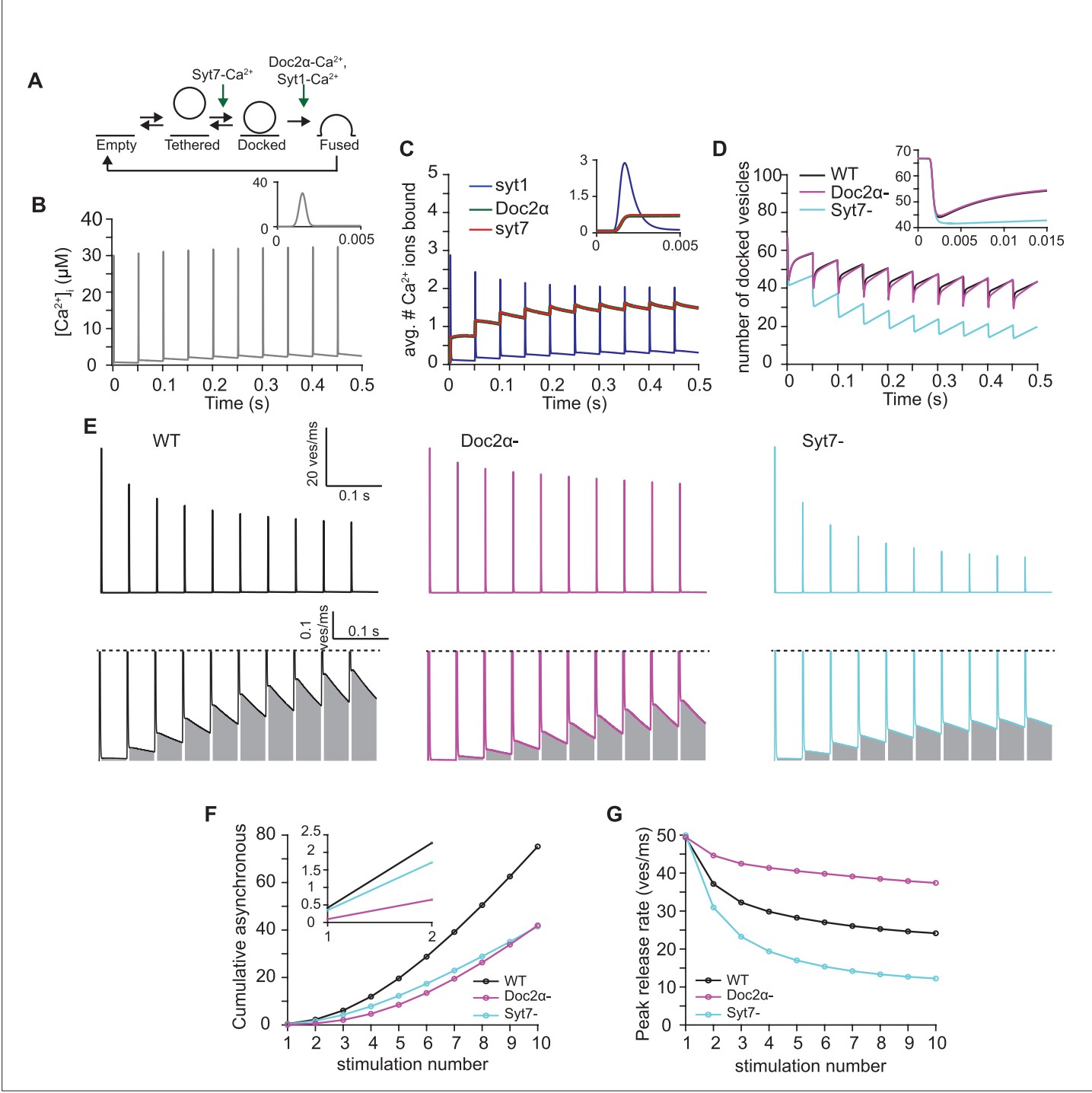

**Figure 7.** A mathematical model in which syt7 catalyzes synaptic vesicle docking and Doc2α promotes fusion that can qualitatively explain the observed phenotypes. (**A**) Simplified scheme of the model in which syt7 acts as a catalyst of synaptic vesicle docking (increasing both docking and undocking rates) and Doc2α regulates fusion together with syt1. All proteins only exert their effect when bound to $Ca^{2+}$. In the model, a release site can either be empty or occupied by a tethered or a docked vesicle. For full model scheme, see ***Figure 7—figure supplement 1***. (**B**) $Ca^{2+}$ transient used to perform the simulations. The transient corresponds to a signal induced by 10 action potentials (APs) at 20 Hz. Inset shows a zoom-in over the first 5 ms of the transient. (**C**) Average number of $Ca^{2+}$ ions bound to syt1, Doc2α, and syt7 over time. All syt1s expressed per synaptic vesicle can maximally bind five $Ca^{2+}$ ions in total, Doc2α and syt7 can each bind two $Ca^{2+}$ ions per release site. Inset shows zoom-in over the first 5 ms of the simulation. (**D**) The number of docked vesicles over time for simulations including functionality of all proteins (black, WT), simulations lacking Doc2α (magenta, Doc2α-), and simulations lacking syt7 (cyan, syt7-). Inset shows a zoom-in over the first 15 ms of the simulation. (**E**) Simulated release rates obtained using the full

*Figure 7 continued on next page*

*Figure 7 continued*

model (WT, left, black), the model lacking Doc2α (Doc2α -, middle, magenta), and the model lacking syt7 (syt7, right, cyan) simulations. Top line shows the entire trace. Bottom panels show a zoom-in to illustrate the asynchronous release component, which is indicated by the gray area. (**F**) Cumulative number of asynchronously released vesicles per stimulus. Asynchronous release is quantified as the release between 5 ms after the start of the AP and the start of the next AP. Inset shows zoom-in over the first two pulses. (**G**) Peak release rates per stimulus for stimulations using the full model (WT, black), the model lacking Doc2α (Doc2α-, magenta), and the model lacking syt7 (syt7-, cyan).

The online version of this article includes the following source data, source code, and figure supplement(s) for figure 7:

**Source data 1.** Data used to generate the graphs in *Figure 7*.

**Source code 1.** Full code used for the mathematical model and simulations.

**Figure supplement 1.** Full reaction scheme of the model.

**Figure supplement 2.** Effects of different model parameters on model predictions.

**Figure supplement 3.** Covarying two parameters illustrates that the selected model parameters are unlikely to provide a unique solution.

**Figure supplement 4.** Evaluation of a model lacking both Doc2α and syt7.

the number of adjusted parameters in the model and ensured that syt7 and Doc2α have the same properties except for which energy barrier they exert their effects (docking vs. fusion). This allowed us to more specifically investigate whether the differences in the reactions on which syt7 and Doc2α act can explain the differences in the apparent roles these two proteins have in AR. Additionally, we assumed that the rates of undocking and untethering are equal to the rates of docking and tethering, respectively. This assumption was made to simplify parameter evaluation of the model. The remaining reaction rates and the effect syt7 and Doc2α exert on the energy barriers were manually adjusted to obtain good qualitative correspondence to the experimental data (see 'Methods', *Table 1*, and *Figure 7—figure supplement 2*). Please note that the selected parameter values do not provide a unique solution (as indicated by parameter space evaluation, *Figure 7—figure supplement 3*), meaning that our modeling effort does not provide reliable estimates of the parameters.

With these parameters and assumptions, we evaluated the model during a stimulus train. We used estimated $Ca^{2+}$ transients resulting from 20 Hz AP trains to drive our model (*Figure 7B*) and monitored $Ca^{2+}$-binding to Doc2α, syt7, and syt1, the number of docked vesicles, the peak rate of synchronous release, and the magnitude of AR. Our simulations indicate that all three sensors reach maximum $Ca^{2+}$ activation at a similar time (*Figure 7C*), despite slower $Ca^{2+}$-binding rates of syt7 and Doc2α compared to syt1. However, syt1's low $Ca^{2+}$ affinity but fast binding kinetics ensure that its $Ca^{2+}$ association closely follows the shape of the $Ca^{2+}$ transient itself (*Figure 7C*). The activation of syt1 by $Ca^{2+}$ dominated the synchronous release observed immediately after the first AP as removal of syt7 and Doc2α from the model did not affect the peak release rate (*Figure 7G*). This only holds if Doc2α has an ~40% lower contribution to reducing the fusion barrier upon $Ca^{2+}$ binding compared to syt1 (*Figure 7G*, *Figure 7—figure supplement 2*). By contrast, AR in response to a single AP strongly depended on the delayed and sustained activation of Doc2α (*Figure 7C F*). Although Doc2α and syt7 have the same $Ca^{2+}$-binding dynamics in the model, syt7 did not contribute noticeably to neurotransmitter release induced by the first AP (*Figure 7F and G*) due to its assigned role in docking. These results from the first action potential during train stimulation are consistent with the experiments (*Figures 1* and *2*).

During the stimulus train in simulations of the control condition, synchronous release gradually decreased, while AR showed the opposite trend (*Figure 7F and G*). This behavior is influenced by the competition between $Ca^{2+}$-bound syt1 and Doc2α for a common resource to elicit their effect on the fusion barrier (*Figure 7C*). Removal of Doc2α from the model caused a reduced depression of synchronous release and reduced AR (compare *Figure 7E and G* with *Figures 3D and 4B*). These results are consistent with the experiments (*Figures 3 and 4*) and suggest that, though less efficient in promoting fusion, Doc2α-mediated fusion has increasingly more influence on synaptic transmission during extended AP trains.

As in the experiments, the physiological relevance of syt7's ability to catalyze vesicle docking was revealed only from the second stimulus onward. Owing to its higher $Ca^{2+}$ affinity and release site localization, syt7, like Doc2α, became increasingly activated over the duration of the stimulus train (*Figure 7C*). $Ca^{2+}$-bound syt7 ensured a fast recovery of docked vesicles after depletion (independent of whether fusion depended on Doc2α and Syt1 or Syt1 alone, *Figure 7D*). As observed experimentally in the syt7 KO (*Figure 6*), removal of this catalytic function in the model resulted in very

limited recovery of docked vesicles before the next stimulus (*Figure 7D*). Because under prolonged stimulation both synchronous and asynchronous release depend on vesicle replenishment, removal of syt7 from the model resulted in the reduction of both release modes (*Figure 7E and G*). This is in agreement with the syt7 KO data (*Figures 3 and 4*). Thus, there is a good qualitative correspondence of our models to the experimental data. This demonstrates that the results of our experiments are consistent with a sequential role of syt7 and Doc2α in promoting synaptic vesicle docking and fusion, respectively.

## Epistasis experiments indicate that syt7 and Doc2α act sequentially

The sequential model predicts that syt7 works upstream of Doc2α to support AR. If syt7 and Doc2α act as parallel $Ca^{2+}$ sensors triggering AR directly, rather than in sequence through transient docking (syt7) and fusion (Doc2α), deleting both proteins would be expected to reduce or eliminate the remaining AR from either of the single knockouts. To test this prediction, we performed an epistasis experiment using DKO mice. For these experiments, we conducted hippocampal slice electrophysiology. We first applied single action potentials and observed that loss of both proteins was not additive with respect to AR: the reduction in slow EPSC charge was identical in Doc2α neurons compared to Doc2α/syt7 DKOs (*Figure 8A and B*). This was the expected result as syt7 single KO had no effect on responses to single action potentials. We then applied train stimulation (*Figure 8C*) and again quantified the tonic charge transfer component as a measure of AR (*Figure 8D*). Importantly, no differences in AR were observed between the Doc2α KO and the Doc2α/syt7 DKO condition. However, the DKO showed a different short-term plasticity phenotype than either single KO. Synchronous release early in the train was similar in DKOs to EGTA-treated neurons, with abolished paired-pulse facilitation but not the enhanced depression seen in syt7 KOs (*Figure 8—figure supplement 1*). This is in line with the model and other experiments showing competition between synchronous and asynchronous release, such that deleting Doc2α slows depression. Note that in EGTA-treated neurons, like in the DKO, both residual $Ca^{2+}$-triggered docking and AR are reduced (*Kusick et al., 2020*), explaining why the DKO short-term plasticity phenotype is so similar to the effects of EGTA. However, in the DKO, late in the train amplitudes decayed to the same lower level as syt7 KOs, suggesting a specific role for syt7 in limiting steady-state depression. Together, these findings strongly support a sequential model for the action of syt7 and Doc2α (illustrated in *Figure 9*, as detailed below).

Consistent with the experimental data, removal of both syt7 and Doc2α from the model resulted in less depression of synchronous release compared to the simulations lacking syt7 only (*Figure 7— figure supplement 4*). This is because (1) $Ca^{2+}$-bound syt1 gains more access to the fusion machinery in this phase when not competing with Doc2α and (2) the reduced AR (in both Doc2αKO and DKO) causes more vesicles to be available for synchronous transmission when they are 'spared' from AR. This matches the experimental observations (*Figure 8—figure supplement 1*). At the same time, the model shows reduced AR after 10 action potentials in simulations lacking both syt7 and Doc2α, compared to simulations in which only one of these proteins had been removed (*Figure 7—figure supplement 4C*). This is in contrast to the experimental data, which detected no difference in the cumulative charge of DKOs compared to Doc2α KOs after 50 action potentials (*Figure 8D*). This could be due to some simplifications that were made while constructing the model such as (1) the implicit simulation of competition between Doc2α and syt1, (2) neglecting potential $Ca^{2+}$-independent roles of these proteins (see 'Discussion'), or (3) the fact that Doc2α is the only slow $Ca^{2+}$ sensor promoting fusion in the model, while the remaining AR in the DKO experiments points to another unidentified sensor. Nonetheless, our modeling confirms that the sequential action of syt7 and Doc2α in the $Ca^{2+}$-dependent docking and fusion of synaptic vesicles, respectively, can explain the phenotypes observed in Doc2α and syt7 KO neurons and partially explain the phenotype of the DKO neurons.

## Discussion
### Resolving the asynchronous release $Ca^{2+}$ sensor controversy

Both Doc2α (*Yao et al., 2011*) and syt7 (*Wen et al., 2010*) have been implicated in AR, but conflicting results over the years led to an unresolved debate – which sensor mediates AR? Several factors may have contributed to these inconsistent results, including stimulation frequency, the type of synapse under study, and method used to measure release. To minimize the effects from these different

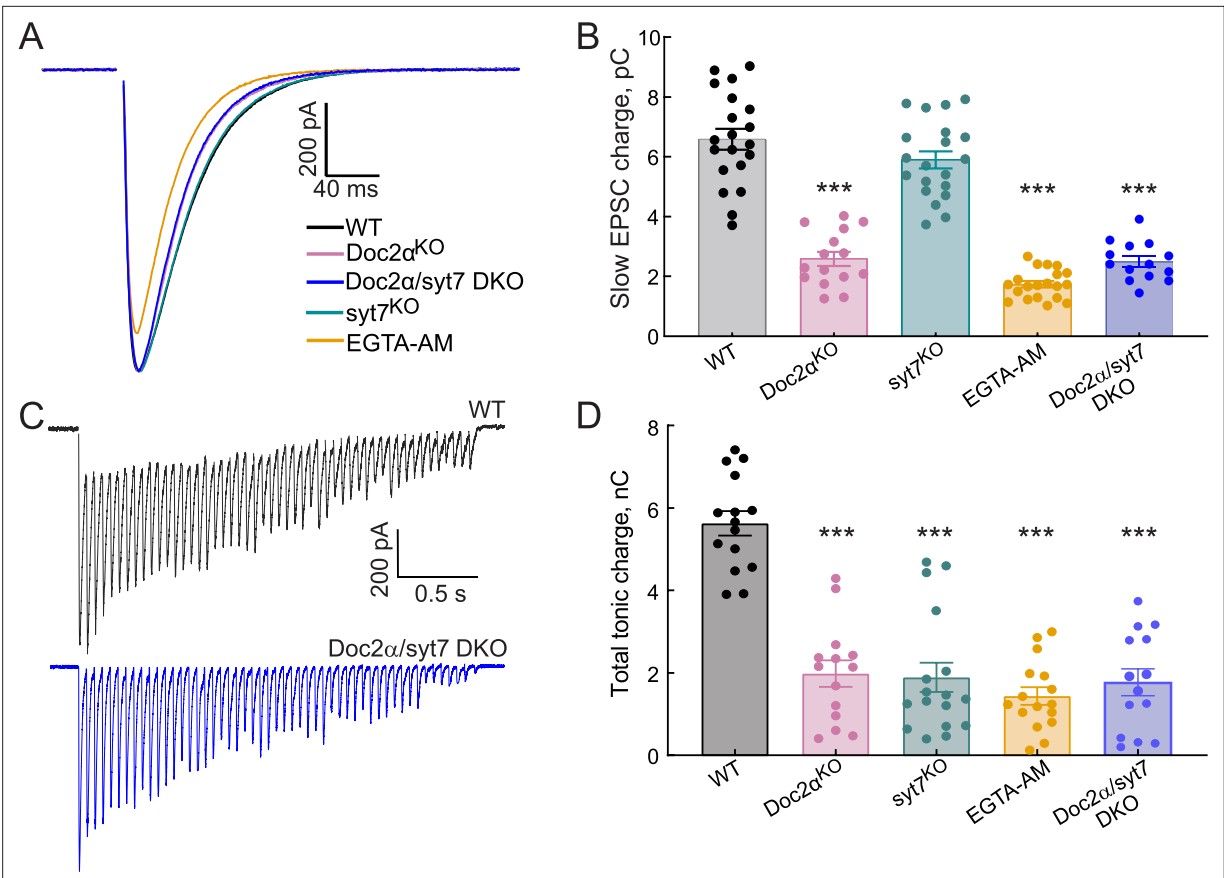

**Figure 8.** Asynchronous release (AR) in Doc2α/syt7 double knockout (DKO) neurons: non-additive effects on AR indicate function through a common pathway. All recordings were carried out using acute hippocampal slices, in CA1, as described in *Figures 2 and 4*. Data from all conditions except Doc2α/syt7 DKO are the same as shown in *Figures 2 and 4* and. (**A**) Averaged traces of evoked excitatory postsynaptic currents (EPSCs) recorded from WT (n = 18 recordings; from five independent litters), Doc2αKO (n = 15; from four independent litters), Doc2α/syt7 DKO (n = 14; from four independent litters), and EGTA-AM-treated neurons (n = 20; from four independent litters). (**B**) Bar graph summarizing the slow EPSC charge values. Error bars are mean ± the standard error of the mean. One-way ANOVA, followed by Dunnett's test, was used to compare mutants against WT. ***p<0.001. Exact p-value for slow EPSC charge in syt7 KO = 0.43. (**C**) Representative EPSCs triggered by 20 Hz stimulus trains using WT and Doc2α/syt7 DKO (n = 15 recordings; from five independent litters) neurons. (**D**) AR in WT, Doc2αKO, syt7KO, Doc2α/syt7 DKO, and EGTA-AM-treated neurons was estimated by measuring the total tonic charge transfer during the 20 Hz stimulus train, as described in *Figure 4C*. Error bars are mean ± the standard error of the mean. One-way ANOVA, followed by Dunnett's test, was used to compare mutants against WT. ***p<0.001. All data, summary statistics, and p-values are listed in *Figure 8—source data 1*.

The online version of this article includes the following source data and figure supplement(s) for figure 8:

**Source data 1.** Data used to generate the graphs in *Figure 8*.

**Figure supplement 1.** EGTA-AM phenocopies synaptic depression of phasic responses in Doc2α/syt7 double knockouts (DKOs), an intermediate phenotype between the single KOs.

**Figure supplement 1—source data 1.** Data used to generate *Figure 8*.

factors, we conducted a side-by-side comparison of Doc2α and syt7 KO neurons using direct optical measurements of glutamate release, patch-clamp electrophysiology, and ultrastructural characterization of millisecond-by-millisecond vesicle dynamics from mouse hippocampal neurons. These experiments show that Doc2α drives significant levels of AR after a single action potential. In sharp contrast, knocking out syt7 has a negligible effect on this mode of release in response to a single action potential. Remarkably, during a stimulus train, both Doc2α and syt7 are equally important for AR.

These results resolve some of the inconsistencies in the literature. First, Doc2α is a Ca$^{2+}$ sensor for AR both after a single stimulus and also during stimulus trains in both cultured hippocampal neurons and Schaffer collaterals. Second, syt7 contributes to AR, but its role is only apparent from the second action potential onward. Thus, at least in the excitatory mouse hippocampal synapses studied here,

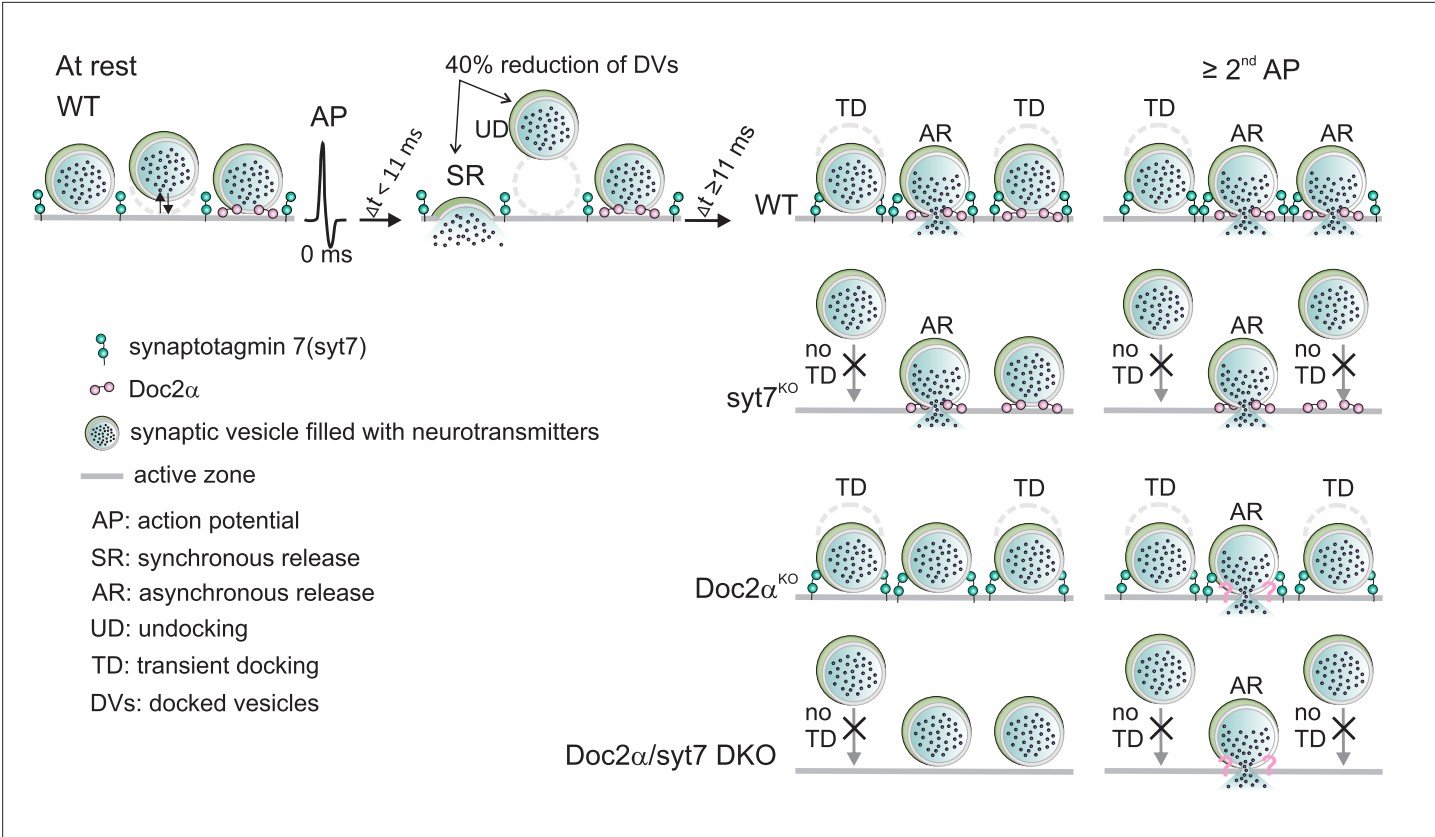

**Figure 9.** Doc2α triggers asynchronous release (AR) in response to single action potentials (APs). During repetitive stimulation, syt7 mediates transient docking to feed docked vesicles to Doc2α for ongoing AR. At rest, synaptic vesicles are docked at the active zone (solid gray line) in a dynamic equilibrium indicated by the antiparallel black arrows (far-left panel). Under these conditions, docking is not affected by syt7 (*Figure 6E*; *Vevea et al., 2021*). A single AP triggers both undocking (UD; *Kusick et al., 2020*) and synchronous release (SR), resulting in a 40% reduction in docked vesicles in ~5 ms (*Figure 6E and F*; *Kusick et al., 2020*). This is followed by a transient docking (TD) step in which vesicles reattach to the active zone via a process that takes ~14 ms and is mediated by Ca²⁺ and syt7 (granite green) (*Figure 6E and F*; *Kusick et al., 2020*). The same single AP also triggers AR mediated directly by Doc2α (lilac), at the 11 ms time point (*Figure 5F*). Syt7 plays a minimal role in release during the first AP (*Figures 1, 2, and 5E*; and references in the main text), again because it does not regulate docking in the resting state. However, the loss of syt7 abolishes activity-dependent TD (*Figure 6E*), which decreases the availability of docked vesicles during subsequent (i.e., two or more) APs, indirectly reducing Doc2α-triggered AR during ongoing activity. As a test of this model, an epistasis experiment revealed that the Doc2α/syt7 double knockout (DKO) AR phenotype was nonadditive (*Figure 8*). This validates a sequential, two-step, model, in which – during ongoing activity – syt7 feeds docked vesicles to Doc2α, which directly mediates AR. Hence, syt7 acts upstream of Doc2α, as a docking factor, while Doc2α functions as a proximal Ca²⁺ sensor for AR in hippocampal synapses. This model can account for why Doc2α, but not syt7, drives a large portion of AR in response to a single AP, while Doc2α and syt7 both contribute to AR during subsequent APs.

these sensors both contribute to AR. However, it is still likely that which of the sensors is required depends on synapse type, since Doc2α played no apparent role in transmission at Purkinje cell to deep cerebellar nuclei synapses (*Khan and Regehr, 2020*). This, along with our finding that syt7/Doc2α DKOs still had remaining AR, indicates that there are other Ca²⁺ sensors for AR. Synaptotagmin 3, a faster high-affinity sensor, was recently proposed to support facilitation and resistance to depression by recruiting docked synaptic vesicles, in much the same manner we described here for syt7 (*Weingarten et al., 2022*). It stands to reason that syt3 could also similarly support AR during repetitive activity. There are likely also sensors yet to be identified that, like Doc2, directly drive AR. Thus, further work should explore the diversity of synaptic Ca²⁺ sensors and how they contribute to heterogeneity in synaptic transmission throughout the brain.

## Physiological relevance of rapid activity-dependent docking triggered by syt7

How syt7 contributes to three separate phenomena – paired-pulse facilitation, AR, and resistance to synaptic depression – is a major question in the field (*Huson and Regehr, 2020*; *Jackman and Regehr, 2017*; *Liu et al., 2014*; *Vevea et al., 2021*). Our data suggest that syt7 drives activity-dependent transient docking to support all three processes. In the case of paired-pulse facilitation, it can supply docked vesicles for syt1-mediated synchronous release; it likely functions in the same manner to reduce synaptic depression during train stimulation. In the case of AR, syt-7-mediated docked vesicles can be used by Doc2α, which then directly triggers this slow mode of transmission. This is consistent with the recent proposal that activity-dependent docking underlies both short-term plasticity and AR (*Kobbersmed et al., 2020*; *Miki et al., 2018*; *Miki et al., 2016*; *Neher and Brose, 2018*; *Silva et al., 2021*; though see *Mendonça et al., 2022* for a correlation between short-term plasticity and AR with a different proposed mechanism).

Three pieces of data reported here support this notion. First and foremost, transient docking, which appears to mirror the time course of paired-pulse facilitation (*Kusick et al., 2020*), is completely blocked in syt7 KO hippocampal neurons. Consequently, these synapses do not facilitate, instead depressing quickly and profoundly, suggesting that syt7-mediated docking is at least in part necessary for facilitation. Second, despite release probability not changing, docked vesicles are more severely depleted (~33% compared to wild type at 5 ms after an action potential) in syt7 KO both after single action potentials (this study) and high-frequency trains (*Vevea et al., 2021*). This implies that vesicles are normally recruited by syt7 to the active zone continuously during the first 15 ms after stimulation in WT synapses. Third, in the absence of the asynchronous $Ca^{2+}$ sensor Doc2α, the recovery of docked vesicles is significantly faster, nearly returning to baseline by 11 ms, suggesting that some vesicles docked by syt7 are fused by Doc2α. Although we cannot rule out the possibility that Doc2α works by docking and then fusing vesicles that are initially undocked, slowed train depression of synchronous release and faster docked vesicle replenishment in Doc2α KO suggests vesicles are still being recruited in the absence of Doc2α, but are less likely to fuse asynchronously. Thus, these data show that syt7-mediated docking contributes to both AR and short-term plasticity. We tested our model for the sequential action of syt7 and Doc2α by conducting an epistasis experiment using DKO mice. Disruption of syt7 did not exacerbate the loss of AR in Doc2α KOs either after single stimuli or train stimulation. These findings are consistent with the idea that syt7 functions upstream of Doc2, with the upstream step (transient docking) required only during sustained activity (summarized in the cartoon model shown in *Figure 9*).

There are important questions posed by this sequential idea. First, while we found no statistically significant effect of syt7 KO on responses to a single action potential in both the hippocampal preparations used in this study, we previously found a very small (2%) but statistically significant decrease in AR in the KO (*Vevea et al., 2021*). By contrast, syt7 is required for the majority of AR after single action potentials at cerebellar granule cell synapses (*Turecek and Regehr, 2018*). Why does syt7 sometimes contribute substantially to responses to single stimuli: is this because transient docking is more important for initial responses at some synapses than others, or is syt7 actually able to trigger synaptic vesicle fusion? Second, the biochemical basis for the different functions of Doc2 and syt7 is unknown: what distinguishes the ability of a $Ca^{2+}$ sensor to trigger docking from its ability to trigger fusion? Curiously, syt7, unlike the fast syts and like the other apparent 'docking sensor' syt3 (*Weingarten et al., 2022*), is strongly enriched not on synaptic vesicles but at the plasma membrane (*Liu et al., 2014*; *Vevea et al., 2021*). In fact, it is nonfunctional when directed to synaptic vesicles (*Vevea et al., 2021*). This is in contrast to syt7's actions outside the synapse and in non-neuronal cells, where it localizes to dense-core vesicles and lysosomes and seems to trigger exocytosis directly (*Chakrabarti et al., 2003*; *Tawfik et al., 2021*; *van Westen et al., 2021*). These are all hints that targeting to the plasma membrane may be relevant to a role in $Ca^{2+}$-triggered docking as opposed to fusion. Still, the mechanism by which syt7 helps promote vesicle docking in response to $Ca^{2+}$ remains to be discovered.

## A sequential model of syt7 and Doc2 in transmitter release

We confirmed that the dynamics observed in the experimental data could be qualitatively recapitulated by a mathematical model in which syt7 and Doc2α act to promote synaptic vesicle docking and fusion, respectively. Both experiments and modeling showed (1) that syt7-mediated docking promotes

synchronous and AR from the second stimulus onward, and (2) that Doc2α promotes AR at the cost of synchronous release. In this work we presented one way to implement the theory. We cannot exclude the possibility that other implementations can also explain the dynamics of the number of docked vesicles and synaptic fusion events observed in the experimental data.

In our model, syt7 acts as a catalyst for synaptic vesicle docking, meaning that Ca²⁺-bound syt7 enhances both docking and undocking rates. This ensures that the number of docked vesicles at rest is independent of the presence of syt7, accounting for the observation that steady-state docking is normal in syt7 KO neurons (*Vevea et al., 2021*; this study). If syt7 functions as a docking catalyst, the number of docked vesicles cannot overshoot during repetitive stimulation, but only return to baseline. While this is consistent with the EM data presented here, it does not explain the established role of syt7 in facilitation (*Jackman et al., 2016*) as facilitation has been proposed to depend on an *increase* in the number of docked/primed vesicles (*Jusyte et al., 2023*; *Kobbersmed et al., 2020*; *Miki et al., 2016*; *Neher and Brose, 2018*). Therefore, it remains possible that syt7 selectively alters docking or undocking rates. Corresponding to the role of syt7 in our model, another model previously showed that syt3 promotes Ca²⁺-dependent docking (*Weingarten et al., 2022*). As in our model, as well as various other models (*Miki et al., 2016*; *Pan and Zucker, 2009*; *Pulido and Marty, 2018*; *Walter et al., 2013*), refilling of the readily releasable pool of vesicles occurred in two steps: vesicles first enter a state from which no fusion can occur (the tethered state in the model presented here). Subsequently, these vesicles become fusion competent (the docking reaction in the model presented here). It is also possible that some of syt7's role in making vesicles fusion-competent is not observable by docking in our electron microscopy experiments. At the moment, it is difficult to know since the cultured hippocampal neurons we used only facilitate very slightly (*Figure 3*). Future studies should prioritize zap-and-freeze experiments in intact circuits on synapses with high paired-pulse ratios to test if an overshoot in docking is actually responsible for synaptic facilitation.

Synaptic vesicle exocytosis in the model was simulated following the dual-sensor model (*Kobbersmed et al., 2020*; *Sun et al., 2007*), in which Ca²⁺ binds to two different sensors promoting fusion (Doc2α and syt1 in the model presented here). We diverged from the dual-sensor model by implementing a competition between syt1 and Doc2α for Ca²⁺ binding. This competition was needed to recapitulate the reduced depression observed in Doc2αKO neurons compared to WT controls (*Figures 3 and 4*). The direct competition between Doc2α and syt1 could indicate a need to act at a limited number of 'slots' at the release site (e.g., SNARE complexes or lipid clusters; *Chen et al., 2021*; *Groffen et al., 2010*; *Honigmann et al., 2013*; *Kobbersmed et al., 2022*; *Yao et al., 2011*; *Zhou et al., 2015*). Additionally, the apparent competition between AR and synchronous release over a limited supply of docked vesicles (as proposed by *Otsu et al., 2004* and *Yao et al., 2011*, and suggested by the EM here) may play a role in the slower depression of synchronous release observed in Doc2α KOs. Which of these two forms of competition between synchronous and asynchronous release, over vesicles or over slots/molecules, is more important at a given synapse likely depends on (1) how limited the supply of docked vesicles is and (2) the relative copy numbers of Doc2α/syt1 in release sites. Our model describes the association to 'slots' implicitly by lumping it together with Ca²⁺ binding and synaptic vesicle fusion. This also means that our model can only account for Ca²⁺-dependent functions of the included proteins. Recent analysis, however, indicates that syt1 pre-associates with these slots at resting Ca²⁺ concentrations, which would limit access to these slots for other sensors (*Kobbersmed et al., 2022*). This arrangement would account for the reported clamping syt1 exerts on spontaneous transmission (*Courtney et al., 2019*; *Kochubey and Schneggenburger, 2011*; *Schupp et al., 2016*) and AR (*Maximov and Südhof, 2005*; *Nishiki and Augustine, 2004*). Accordingly, there will be situations where the model falls short in providing an explanation. Nonetheless, it shows that sequential action of syt7 and Doc2α can explain the role these proteins have in docking, short-term plasticity, and AR in mouse hippocampal synapses.

# Materials and methods

## Key resources table

| Reagent type (species) or resource | Designation | Source or reference | Identifiers | Additional information |
|---|---|---|---|---|
| Biological sample (*M. musculus*) | Acute mouse hippocampal slices | *Xue et al., 2018* | *C57BL/6-Doc2a^{tm1Ytk};C57BL/6-Syt7^{tm1Nan}* | We are unable to distribute this strain, but a DKO strain can be generated by crossing *C57BL/6-Syt7^{tm1Nan}* (available from The Jackson Laboratory) and *Doc2a^{em1smoc}* (available from the Shanghai Model Organisms Center) or C57BL/6JCya-*Doc2a^{em1}* (available from Cyagen) |
| Biological sample (*M. musculus*) | Acute mouse hippocampal slices | The Jackson Laboratory | *C57BL/6-Syt7^{tm1Nan}* | |
| Biological sample (*M. musculus*) | Acute mouse hippocampal slices | *Sakaguchi et al., 1999; Groffen et al., 2010* | *C57BL/6-Doc2a^{tm1Ytk}* | Mouse line provided by the Verhage lab. We are unable to distribute this strain, and it is not available from a repository, but an almost-identical allele is available from the Shanghai Model Organisms Center: *Doc2a^{em1smoc}*, and from Cyagen: C57BL/6JCya-*Doc2a^{em1}*. |
| Biological sample (*M. musculus*) | Primary mouse hippocampal neurons | The Jackson Laboratory | *C57BL/6-Syt7^{tm1Nan}* | |
| Biological sample (*M. musculus*) | Primary mouse hippocampal neurons | *Sakaguchi et al., 1999; Groffen et al., 2010* | *C57BL/6-Doc2a^{tm1Ytk}* | Mouse line provided by the Verhage lab. We are unable to distribute this strain, and it is not available from a repository, but an almost-identical allele is available from the Shanghai Model Organisms Center: *Doc2a^{em1smoc}*, and from Cyagen: C57BL/6JCya-*Doc2a^{em1}*. |
| Recombinant DNA reagent | pF(UG) CamKII sf iGluSnFR A184V | *Vevea and Chapman, 2020*; RRID:Addgene#158769 | | |
| Recombinant DNA reagent | pF(UG) CamKII sf iGluSnFR S72A | *Vevea et al., 2021*; RRID:Addgene#179726 | | |
| Chemical compound, drug | EGTA-AM | Millipore #324628 | | |
| Software, algorithm | MATLAB 2021: SynapsEM code | *Watanabe et al., 2020* | | |
| Software, algorithm | Prism 9 | | | |
| Software, algorithm | Fiji: SynapsEM code | *Watanabe et al., 2020* | | |
| Software, algorithm | MATLAB 2021: model code | This paper | | See *Figure 7—source code 1* |

All animal care was performed according to the National Institutes of Health guidelines for animal research with approval from the Animal Care and Use Committees at the Johns Hopkins University School of Medicine and the University of Wisconsin, Madison (assurance number: A3368-01). Both male and female syt7^{KO}, Doc2α^{KO}, and WT littermates were used.

Mouse lines used for this study were as follows:

C57BL/6-*Syt7^{tm1Nan}* (*Chakrabarti et al., 2003*)
C57BL/6-*Doc2a^{tm1Ytk}* (*Sakaguchi et al., 1999*)
*C57BL/6-Doc2a^{tm1Ytk}*; C57BL/6-*Syt7^{tm1Nan}*

Double knockouts were maintained as homozygous null. Single knockout lines were maintained as heterozygotes.

## Neuronal cell culture

For single-knockout experiments, Doc2α and syt7 KO mice were maintained as heterozygotes so that littermate controls could be used. These mice were bred, and postnatal day 0–1 pups were geno-typed, with homozygotes for the null allele (–/–) and the wild-type allele (+/+) being used for culture. For DKO experiments, mice were maintained as homozygotes, with *Doc2α^{+/+}* used as the wild-type control. For iGluSnFR experiments, WT and KO hippocampal neurons were grown in mass culture for 14 DIV. In brief, hippocampi were dissected from mouse brains and digested for 25 min at 37°C in 0.25% trypsin-EDTA (Corning). After mechanical dissociation, hippocampal neurons were plated on 12 mm glass coverslips pretreated with poly-D-lysine (Thermo Fisher). Neurons were cultured in Neurobasal A (Gibco)-based media with B27 (2%, Thermo Fisher) and GlutaMAX (2 mM, Gibco) at 37°C with 5% $CO_2$.

For high-pressure freezing and electron microscopy, cell cultures were prepared on 6 mm sapphire disks atop a feeder layer of astrocytes, as previously described (*Kusick et al., 2020*). Astrocytes were harvested from cortices with the treatment of trypsin (0.05%) for 20 min at 37°C, followed by trituration and seeding on T-75 flasks containing DMEM supplemented with 10% FBS and 0.2% penicillin-streptomycin. After 2 wk, astrocytes were plated on 6 mm sapphire disks (Technotrade Inc) coated with poly-D-lysine (1 mg/ml), collagen (0.6 mg/ml), and 17 mM acetic acid at a density of $13 \times 10^3$ cells/cm$^2$. Genotyping was performed either (1) after hippocampal dissection, using cortices lysed with 10 mM Tris-HCL pH 8.0, and 100 mM NaCl in the presence of 50 µg proteinase K, with hippocampi either left in Neurobasal A at 37°C before switching to papain after or left in dissection medium (1% penicillin-streptomycin, 1 mM pyruvate, 10 mM HEPES, 3 mM glucose in HBSS) at 4°C, or (2) using tail clips (alkaline lysis; *Truett et al., 2000*) of live mice, which were then sacrificed and dissected after genotyping was finished. We note that leaving the neurons in dissection medium at 4°C results in more consistent cell health than Neurobasal A at 37°C, particularly when hippocampi are left for several hours while genotyping is performed. Hippocampi were dissected under a microscope and digested with (1) papain (0.5 mg/ml) and DNase (0.01%) in the dissection medium for 25 min at 37°C with gentle perturbation every 5 min or (2) papain (0.5 mg/ml) in Neurobasal A with 200 rpm shaking. After trituration, neurons were seeded onto astrocyte feeder layers at a density of $20 \times 10^3$ cells/cm$^2$ (75,000 per well) and cultured in Neurobasal A with 2% B27 and 2 mM at 37°C with 5% CO$_2$. Before use, sapphire disks were carbon-coated with a '4' to indicate the side that cells are cultured on. The health of the cells, as indicated by de-adhered processes, floating dead cells, and excessive clumping of cell bodies, was assessed immediately before experiments.

## Plasmids

pF(UG) CamKII sf iGluSnFR A184V (*Vevea and Chapman, 2020*); Addgene #158769.
pF(UG) CamKII sf iGluSnFR S72A (*Vevea et al., 2021*); Addgene #179726.

## Lentivirus production and use

For viral transduction, the original (*Marvin et al., 2013*) and low-affinity S72A (*Marvin et al., 2018*) iGluSnFR variants were used. iGluSnFRs were subcloned into a FUGW transfer plasmid (*Vevea and Chapman, 2020*). Lentiviral particles were generated as previously described (*Vevea et al., 2021*; *Vevea and Chapman, 2020*) by co-transfecting the transfer, packaging, and helper (pCD/NL-BH*ΔΔΔ and VSV-G encoding pLTR-G) plasmids into HEK 293/T cells. Lentivirus was collected from the media 48–72 hr after transfection and concentrated by ultracentrifugation. Viruses were titrated and then used to infect neurons on DIV 3–5.

## iGluSnFR imaging and quantification

iGluSnFR fluorescence intensity changes were recorded and analyzed as previously described (*Vevea et al., 2021*; *Vevea and Chapman, 2020*), with slight adjustments to imaging time and binning. To measure iGluSnFR signals in response to a single stimulus, the medium-affinity iGluSnFR (A184V) was used in *Figure 1A–D* and *Figure 1—figure supplement 1*. For train stimulation, the low-affinity iGluSnFR sensor (S72A) was used because it more effectively resolved individual responses from high-frequency stimulation due to its faster decay time (*Vevea et al., 2021*). Imaging was performed in media (305 mOsm) comprising (in mM) 128 NaCl, 5 KCl, 2 CaCl$_2$, 1 MgCl$_2$, 30 D-glucose, and 25 HEPES, pH 7.3, with D-AP5 (50 µM) (Abcam; ab120003), CNQX (20 µM) (Abcam; ab120044), picrotoxin (100 µM) (Tocris; 1128) added to prevent recurrent excitation. Single-image planes were acquired ($2 \times 2$ binning) with 10 ms exposure (100 Hz) using 488 nm excitation and 520 emission. Single planes were selected so that the highest amount of iGluSnFR-expressing dendrite arbor was in focus. Laser intensity was set so that the amplitude of iGluSnFR peaks (ΔF/F0) from a 0.5 Hz field stimulus did not change over the course of at least 20 s (photobleaching error). For single-stimuli imaging, 200 frames (10 ms exposure/2 s total) were collected and a single-field stimulus was triggered at 500 ms after the initial frame. For train stimuli imaging, 50 stimuli at 10 Hz (5 s) or 20 Hz (2.5 s) were applied via field stimulation, and 500 or 750 frames, each with a 10 ms exposure time, were collected. Single and train (10 Hz) stimuli (1 ms square pulses) were triggered by a Grass SD9 stimulator through platinum parallel wires attached to a field stimulation chamber (Warner Instruments; RC-49MFSH). Voltage for

field stimulus was set to the lowest voltage that reliably produced presynaptic $Ca^{2+}$ transients in most (>95%) presynaptic boutons (5–30 V). All iGluSnFR imaging experiments were performed at 33–34°C. Temperature and humidity were controlled during experiments by a Tokai incubation controller and chamber. Glutamate release signals were defined as fluorescence intensity changes that were more than five times the standard deviation of the noise (*Figure 1—figure supplement 1C*). Timing of release was determined based on the frame in which the signal peaked, including for dual peaks in the case of synchronous and asynchronous release at the same bouton. For all iGluSnFR experiments, we initially recorded 500 ms as the baseline; the mean intensity was defined as F0. iGluSnFR fluorescence amplitude changes are reported as ΔF/F0. Throughout this study, we term regions of interest as boutons because these regions were defined as yielding at least one iGluSnFR response during the first three stimuli of a 10 Hz train. iGluSnFR data were analyzed using a custom ImageJ plugin (*Vevea and Chapman, 2020*) as follows: the macro begins by subtracting the average projection intensity (the pre-stimulation baseline) from a maximum intensity projection (the largest intensity from the entire image series). The result was duplicated, and the mean was filtered by an area of a radius of 10 pixels. The copied image is converted to a 32-bit image for subsequent calculations. The original duplicated and filtered images are subtracted, creating a new image ('Result'). Thresholding is applied to the 'Result' image, converting it to a binary image. Watershed segmentation was then used to separate touching objects. The results were copied and imported into Axograph X 1.7.2 (Axograph Scientific), where traces were normalized, and background subtracted using pre-stimulus data. All iGluSnFR imaging and analysis were performed blind to genotype.

## Brain slice preparation

Acute hippocampal slices were prepared from mice aged between P30 and P60. Animals of both sexes were anesthetized using isoflurane and then rapidly decapitated. Brains were quickly removed and placed into ice-cold NMDG- HEPES artificial cerebral spinal fluid (aCSF; 300–310 mOsm) comprising (in mM) 92 N-methyl-D- glucamine (NMDG), 2.5 KCl, 1.25 $NaH_2PO_4$, 30 $NaHCO_3$, 25 glucose, 2 thiourea, 5 Na-ascorbate, 3 Na-pyruvate, 0.5 $CaCl_2$, and 10 $MgSO_4$, and 20 HEPES, pH 7.3. This solution was continuously bubbled with 95% $O_2$ and 5% $CO_2$. Also, 300-μm-thick coronal brain sections containing the hippocampal region were cut using a vibratome (Leica VT1200, Leica, Germany). To avoid any potential recurrent excitation, a sharp cut was made between CA3 and CA1. Slices were placed in a chamber and allowed to recover in 150 ml aCSF at 34°C for 25 min. Sodium (2 M stock in aCSF) was added back at the following time intervals: 0 min (250 μl), 5 min (250 μl), 10 min (500 μl), 15 min (1 ml), and 20 min (2 ml) during the 25 min recovery incubation. Slices were then gently placed into another chamber containing aCSF, but with 92 mM NMDG replaced with 92 mM NaCl, at room temperature (at least 1 hr) before being transferred to the recording chamber.

## Electrophysiological recordings in acute hippocampal slices

Whole-cell voltage-clamp recordings were performed under an upright microscope (BX51WI, Olympus, Melville, NY) equipped with a water-immersion objective (LUMPlanFL N ×40x/0.8 W; Olympus). Hippocampal slices were continuously perfused (2 ml/min) with carbogen-saturated, modified aCSF (in mM: 125 NaCl, 25 $NaHCO_3$, 25 glucose, 2.5 KCl, 2 $CaCl_2$, 1.25 $NaH_2PO_4$, and 1 $MgCl_2$, adjusted to 300–310 mOsm). For perfusion, aCSF was heated to 34°C using a Single Channel Temperature Controller (TC-324C, Warner Instruments, Holliston, MA). Recording pipettes were prepared from borosilicate glass (1.5 mm diameter glass capillary, Sutter Instrument, Novato, CA) using a two-step puller (PP-830, Narishige, Japan) and were polished using an MF-830 micro forge (Narishige) immediately prior to use. Pipette position was controlled using an MP-285 motorized micromanipulator (Sutter Instrument); the resistances were typically 4–6 MΩ when filled with recording solution consisting of (in mM) 135 K- gluconate 5 KCl, 10 HEPES, 1 EGTA, 6 $Na_2$ phosphocreatine, 0.3 $Na_3GTP$, and 2 MgATP. To chelate residual $Ca^{2+}$, EGTA-AM (Millipore, Burlington, MA) was added to the bath at a final concentration of 25 μM for 30 min; neurons were then allowed to recover for another 20 min before being placed in the recording chamber. Neurons in the CA1 region were visualized using a Retiga electro CCD camera (Teledyne Photometrics, Tucson, AZ). Patched neurons were held at −70 mV, with access resistance (Ra) less than 20 MΩ throughout the recordings. A concentric bipolar electrode (FHC, 125/50 μm extended tip) was placed in the stratum radiatum (Schaffer collateral fiber) ~300–500 μm away from the recording pipette (note that this large bore diameter is responsible for

the large currents in our recordings). Stimulation strength was selected by starting at 0.5–1.5 mA, steadily increasing stimulation strength and adjusting electrode position until stable currents from 1 Hz test stimuli were achieved. EPSCs were pharmacologically isolated by bath application of 50 µM D-APV and 20 µM bicuculline. QX-314 chloride (5 mM) was in the pipette solution for evoked recordings. All data were collected in voltage-clamp mode using a Multiclamp 700B patch-clamp amplifier (Molecular Devices, San Jose, CA) and a Digidata 1500B (Axon Digidata 1500B, Molecular Devices). AR during 20 Hz stimulation (i.e., tonic charge transfer during the train) was quantified as illustrated in *Figure 4A*. This was done by calculating the area between the baseline, before the first stimulus, and the tail current values, which were measured 1ms before each stimulus in the train (*Otsu et al., 2004*; *Yao et al., 2011*). All recordings and analysis were performed blind to genotype.

## High-pressure freezing and freeze substitution

Cells cultured on sapphire disks were frozen using an EM ICE high-pressure freezer (Leica Microsystems), exactly as previously described (*Kusick et al., 2020*). The freezing apparatus was assembled on a table heated to 37°C in a climate control box, with all solutions pre-warmed (37°C). Sapphire disks with neurons were carefully transferred from the culture medium to a small culture dish containing physiological saline solution (140 mM NaCl, 2.4 mM KCl, 10 mM HEPES, 10 mM glucose; pH adjusted to 7.3 with NaOH, 300 mOsm). NBQX (3 µM; Tocris) and bicuculline (30 µM; Tocris) were added to the physiological saline solution to block recurrent synaptic activity. $CaCl_2$ and $MgCl_2$ concentrations were 1.2 mM and 3.8 mM, respectively. After freezing, samples were transferred under liquid nitrogen to an EM AFS2 freeze substitution system at –90°C (Leica Microsystems). Using pre-cooled tweezers, samples were quickly transferred to anhydrous acetone at –90°C. After disassembling the freezing apparatus, sapphire disks with cells were quickly moved to cryovials containing freeze substitution solutions. For the first two of three Syt7 knockout freezes, freeze substitution was performed exactly as previously described in *Kusick et al., 2020*: solutions were 1% glutaraldehyde, 1% osmium tetroxide, and 1% water in anhydrous acetone, which had been stored under liquid nitrogen then moved to the AFS2 immediately before use. The freeze substitution program was as follows: –90°C for 6–10 hr (adjusted so that substitution would finish in the morning), 5°C hr⁻¹ to –20°C, 12 hr at –20°C, and 10 °C hr⁻¹ to 20°C. For all other experiments, we used a different freeze substitution protocol that yields more consistent, high-contrast morphology, as described in *Vevea et al., 2021*: samples were first left in 0.1% tannic acid and 1% glutaraldehyde at –90°C for ~36 hr, then washed 5×, once every 30 min, with acetone, and transferred to 2% osmium tetroxide, then run on the following program: 11 hr at –90°C, 5°C hr⁻¹ to –20°C, –20°C for 12 hr, 10°C hr⁻¹ to –4°C, then removed from the freeze substitution chamber and warmed at room temperature for ~15 min before washing. For the latter protocol, instead of cryovials, solutions and sapphires were stored in universal sample containers (Leica Microsystems), covered in aclar film to prevent evaporation. Using these containers instead of cryovials makes all solution transfers, from fixation through infiltration and embedding, much faster and more convenient.

## Infiltration, embedding, sectioning, and transmission electron microscopy

Infiltration, embedding, and sectioning were performed as previously described (*Kusick et al., 2020*), except without en bloc staining, which we found to not improve morphology. Samples in fixatives were washed three times, 10 min each, with anhydrous acetone, then left in 30% epon araldite (epon, 6.2 g; araldite, 4.4 g; DDSA, 12.2 g; and BDMA, 0.8 ml, diluted in acetone) for 3 hr, then 70% epon araldite for 2 hr, both with shaking. Samples were then transferred to caps of polyethylene BEEM capsules (EMS) and left in 90% epon araldite overnight at 4°C. The next morning, samples were transferred to 100% epon araldite for 1 hr, then again to 100% for 1 hr, and finally transferred to 100% epon araldite and baked at 60°C for 48 hr. For ultramicrotomy, 40 nm sections were cut. Sections on single-slot grids coated with 0.7% pioloform were stained with 2.5% uranyl acetate then imaged either at 120 kV on the 57,000× setting on a Thermo Fisher Talos L120C equipped with a Ceta camera (pixel size = 244 pm; first replicate of the Doc2 experiment) or at 80 kV on the 100,000× setting on a Hitachi 7600 TEM equipped with an AMT XR50 camera run on AMT Capture v6 (pixel size = 560 pm, all other experiments). Samples were blinded before imaging. To further limit bias, synapses were found by bidirectional raster scanning along the section at the magnification at which images were taken, which

makes it difficult to 'pick' certain synapses, as a single synapse takes up most of this field of view. Synapses were identified by a vesicle-filled presynaptic bouton and a postsynaptic density. Postsynaptic densities are often subtle in our samples, but synaptic clefts were also identifiable by (1) their characteristic width, (2) the apposed membranes following each other closely, and (3) vesicles near the presynaptic active zone. A total of 125–150 micrographs per sample of anything that appeared to be a synapse were taken without close examination. All images were from different synapses (i.e., images were never captured from different sections of the same synapse). Most of the time all images for a condition were taken from a single section, each of which typically contains hundreds of different boutons; when different sections were used, they were always at least 10 serial sections apart (400 nm in z) to ensure they did not contain any of the same boutons.

## Electron microscopy image analysis

EM image analysis was performed as previously described (*Kusick et al., 2020*). All the images from a single experiment were randomized for analysis as a single pool. Only after this randomization were any images excluded from analysis either because they appeared to not contain a bona fide synapse or the morphology was too poor for reliable annotation. The plasma membrane, the active zone, docked synaptic vesicles, and all synaptic vesicles within ~100 nm of the active zone were annotated in ImageJ using SynapsEM plugins (*Watanabe et al., 2020*; https://github.com/shigekiwatanabe/ SynapsEM, copy archived at *Watanabe, 2024*). The active zone was identified as the region of the presynaptic PM with the features described above for identifying a synapse. Docked vesicles were identified by their membrane appearing to be in contact with the PM at the active zone (0 nm from the PM); that is, there are no lighter pixels between the membranes. Vesicles that were not manually annotated as docked but were 0 nm away from the active zone PM were automatically counted as docked when segmentation was quantitated (see below) for data sets counting the number of docked vesicles. Likewise, vesicles annotated as docked were automatically placed in the 0 nm bin of vesicle distances from the PM. To minimize bias and error, and to maintain consistency, all image segmentation, still in the form of randomized files, was thoroughly checked and edited by a second member of the lab. Features were then quantitated using the SynapsEM family of MATLAB (MathWorks) scripts (https://github.com/shigekiwatanabe/SynapsEM, copy archived at *Watanabe, 2024*). Example electron micrographs shown were adjusted in brightness and contrast to different degrees (depending on the varying brightness and contrast of the raw images), rotated, and cropped in ImageJ before import into Adobe Illustrator.

## Statistics

Statistical analysis was performed using GraphPad Prism 9 software. iGluSnFR and electrophysiology data are presented as mean ± standard error of the mean. Electron microscopy data were presented as mean and 95% confidence interval of the mean. For iGluSnFR and electrophysiology, data sets were tested for normality using the Anderson–Darling test and corresponding parametric or nonparametric hypothesis tests performed, as stated in the figure legends. Since electron microscopy pit and docked vesicle data are count data, following a roughly binomial rather than normal distribution, means were used to graph the data (medians are not a useful representation of central tendency for count data, particularly low-frequency count data; e.g., the median number of pits in all samples is 0 since exocytic pits are rare) and Games–Howell post hoc test used, which does not assume a normal distribution. Sample size and number of biological replicates were not determined by power analysis, but were predetermined based on our prior experience with electrophysiology, iGluSnFR, and zap-and-freeze data.

## Mathematical model

In the model we developed, a release site can be in one of three states: empty, occupied with a tethered vesicle, or occupied with a docked vesicle (for full scheme, see *Figure 7—figure supplement 1*; for full underlying code, see *Figure 7—source code 1*,). Only docked vesicles can fuse and their fusion rates depend on the number of $Ca^{2+}$ ions bound to syt1 and Doc2α. $Ca^{2+}$ binding to these $Ca^{2+}$ sensors followed the dual-sensor scheme as previously published (*Kobbersmed et al., 2020*; *Sun et al., 2007*), in which five $Ca^{2+}$ ions per vesicle can bind to the first $Ca^{2+}$ sensor for fusion (syt1 in our model), and two $Ca^{2+}$ ions to the second $Ca^{2+}$ sensor for fusion (Doc2α in our model). We adjusted

the dual-sensor model such that no more than five Ca²⁺ ions can be bound in total to syt1 and Doc2α simultaneously. This is based on the idea that syt1 and Doc2α are competing for the same resources, which could, for instance, be a limited number of SNARE complexes that are available to execute the fusion reaction. Ca²⁺ can bind to Doc2α at any of the release site states, whereas Ca²⁺ binding to syt1 only occurs when vesicles are docked. This is because Doc2α is assumed to reside on the release site (**Verhage et al., 1997**) and syt1 is attached to the synaptic vesicle (**Takamori et al., 2006**). Together this means that an empty release site can be in three different states that are described by the following differential equations:

$$d\frac{E[0]}{dt} = -k_{tet}E[0] + k_{untet}T[0] + \sum_{i=0}^{5}\left(f(0,i)\,D[0,i]\right) + k_{2-}E[1] - 2[ca^{2+}]k_{2+}E[0]$$

$$d\frac{E[1]}{dt} = -k_{tet}E[1] + k_{untet}T[1] + \sum_{i=0}^{5}\left(f(1,i)\,D[1,i]\right) + 2k_{2-}b_{2}E[2] + 2[ca^{2+}]k_{2+}E[0] - \left(\left[ca^{2+}\right]k_{2+} + \right.$$

$$\left. k_{2-}E[1]\right)$$

$$\frac{E[2]}{dt} = -k_{tet}E[2] + k_{untet}T[2] + \sum_{i=0}^{5}\left(f(2,i)\,D[2,i]\right) + [ca^{2+}]k_{2+}E[1] - 2k_{2-}b_{2}E[2] \tag{1}$$

In these equations, E[d2], T[d2], D[d2, s1] denotes the Ca²⁺-binding state of the empty release, tethered vesicle, and docked vesicle, respectively, with d2 the number of Ca²⁺ ions bound to Doc2α and s1 the number of Ca²⁺ ions bound to syt1 per release site. f is the fusion rate function that is described in **Equation 5**. A definition of the model parameters indicated in the equations can be found in **Table 1**. A release site occupied with a tethered vesicle can be in three different states, depending on the Ca²⁺-binding state of Doc2α.

$$d\frac{T[0]}{dt} = k_{tet}E[0] - k_{untet}T[0] + \sum_{i=0}^{5}\left(undocking\,D[0,i]\right) - docking\,T[0] + k_{2-}T[1] - 2[Ca^{2+}]k_{2+}T[0]$$

$$d\frac{T[1]}{dt} = k_{tet}E[1] - k_{untet}T[1] + \sum_{i=0}^{5}\left(undocking\,D[1,i]\right) - docking\;\;T[1] + 2k_{2-}b_{2}T[2] +$$

$$2[Ca^{2+}]k_{2+}T[0] - ([Ca^{2+}]k_{2+} + k_{2-})T[1]$$

$$d\frac{T[2]}{dt} = k_{tet}E[2] - k_{untet}T[2] + \sum_{i=0}^{5}\left(undocking\,D[2,i]\right) - docking\,T[2] + [Ca^{2+}]k_{2+}T[1] - 2k_{2-}b_{2}T[2] \tag{2}$$

Undocking and docking rates are calculated based on the equations below (**Equations 6 and 7**). The following **Equation 2** describes the Ca²⁺ bound state of docked vesicles:

$$
\begin{aligned}
d\frac{D[d2,s1]}{dt} = \;&\mathbf{1}_{s1\geq1}\,\max(0,5-(d2+s1-1))[Ca^{2+}]k_{1+}\,D[d2,s1-1]\\
&+\mathbf{1}_{s1<5}(s1+1)b_{1}^{s1}k_{1-}D[d2,s1+1]\\
&+\mathbf{1}_{d2\geq1}\,\min(\max(0,2-d2+1),\max(0,5-(d2+s1-1)))\,[Ca^{2+}]k_{2+}\,D[d2-1,s1]\\
&+\mathbf{1}_{d2<2}(d2+1)b_{2}^{d2}\,k_{2-}\,D[d2+1,s1]\\
&+\mathbf{1}_{s1=0}\,docking\,T[d2] - (\max(0,5-(d2+s1))[Ca^{2+}]k_{1+} + s1\,b_{1-}^{s1}\,k_{1-}\\
&+ \min(\max(0,2-d2),\max(0,5-(d2+s1)))\,[Ca^{2+}]k_{2+} + d2\,b_{2}^{d2-1}\,k_{2-}\\
&+f(d2,s1) + undocking)\,D[d2,s1]
\end{aligned}
\tag{3}
$$

Fusion can occur from either of the docked states with f being the fusion rate function (**Equation 5**). The change in the cumulative number of fused vesicles (F) is described by:

$$\frac{dF}{dt} = f(d2,s1)\,D[d2,s1] \tag{4}$$

where f is the fusion rate function:

$$f(d2,s1) = l_{+f_{1}^{s1}f_{2}^{d2}} \tag{5}$$

**Table 1.** Parameters of the mathematical model.

| Parameter name | Description | Value | Value determined by |
|---|---|---|---|
| $N_{sites}$ | Number of available release sites (limits the number of docked/tethered vesicles) | 200 | * (only determines the scaling of the responses) |
| $k_{tet}$ | Tethering rate | 7 s⁻¹ | † |
| $k_{untet}$ | Untethering rate constant | 7 s⁻¹ | * (same as $k_{tet}$) |
| $k_{docking}$ | Docking rate constant | 5 s⁻¹ | † |
| $k_{undocking}$ | Undocking rate constant | 5 s⁻¹ | * (same as $k_{docking}$) |
| $l_+$ | Syt1 and Doc2 independent fusion rate constant/ basal fusion rate | 3.5e-4 s⁻¹ | ‡ *Kochubey and Schneggenburger, 2011* |
| **Parameters Ca2+ transient** | | | |
| Amplitude | Amplitude of synchronous Ca²⁺ transient in response to AP | 30 µM | † |
| Residual Ca²⁺ amplitude | Amplitude of the residual Ca²⁺ signal (following single-exponential decay) | 0.75 µM | † |
| Tau | Decay time constant of residual Ca²⁺ signal | 0.2 s | † |
| $Ca^{2+}_{rest}$ | Resting calcium concentration | 0.05 µM | ‡ *Helmchen et al., 1997* |
| **Parameters syt7** | | | |
| $N_7$ | Ca²⁺ cooperativity of syt7 | 2 | * (same as for Doc2) |
| $K_{D7}$ | Dissociation constant, syt7 | 1.5 µM | ‡ *Brandt et al., 2012* |
| $k_{7+}$ | Ca²⁺-binding rate, syt7 | 28 µM⁻¹ s⁻¹ | * (same as for Doc2) |
| $k_{7-}$ | Ca²⁺-unbinding rate constant, syt7 | 42 s⁻¹ | ‡ (computed using kd) |
| $b_7$ | Cooperativity factor, syt7 | 0.5 | * (same as for Doc2 and syt1) |
| $f_7$ | Effect single Ca²⁺ binding to syt7 has on $k_{docking}$ and $k_{undocking}$ | 10 (1 for syt7-simulations) | † |
| **Parameters Doc2α** | | | |
| $N_2$ | Ca²⁺ cooperativity of Doc2α | 2 | ‡ *Sun et al., 2007* |
| $K_{D2}$ | Dissociation constant, Doc2α | 1.5 µM | * (same as for syt7) |
| $k_{2+}$ | Ca²⁺-binding rate, Doc2α | 28 µM⁻¹ s⁻¹ | ‡ (5× reduction compared to syt1, based on *Yao et al., 2011*; 0 for Doc2α- simulations) |
| $k_{2-}$ | Ca²⁺-unbinding rate constant, Doc2α | 42 s⁻¹ | ‡ (computed using kd) |
| $b_2$ | Cooperativity factor, Doc2α | 0.5 | * (same as for syt1 and syt7) |
| $f_2$ | Effect single Ca²⁺ binding to Doc2α has on the fusion rate | 16 | † |
| **Parameters syt1** | | | |
| $N_1$ | Ca²⁺ cooperativity of Syt1 | 5 | ‡ *Kochubey and Schneggenburger, 2011* |
| $k_{1+}$ | Ca²⁺-binding rate, Syt1 | 140 µM⁻¹ s⁻¹ | ‡ *Kochubey and Schneggenburger, 2011* |
| $k_{1-}$ | Ca²⁺-unbinding rate constant, syt1 | 4000 s⁻¹ | ‡ *Kochubey and Schneggenburger, 2011* |
| $b_1$ | Cooperativity factor, syt1 | 0.5 | ‡ *Kochubey and Schneggenburger, 2011* |
| $f_1$ | Effect single Ca²⁺ binding to syt1 has on the fusion rate | 27.98 | ‡ *Kochubey and Schneggenburger, 2011* |

*Set based on additional model assumptions or simplifications.
†Adjusted to match experimental data.
‡Based on literature.

$f_1$ and $f_2$ are the factors by which each Ca²⁺ ion bound to either syt1 (s1) or Doc2α (d2) increases the release rates of a synaptic vesicle, and $l_+$ equals the basal fusion rate (*Table 1*). In thxe model, docking and undocking rates are dependent on the number of Ca²⁺ ions bound to syt7 (s7). Binding of Ca²⁺ to all syt7s expressed per release site follows the same scheme as Doc2α and can be described by the following rate equations:

$$d\frac{S[0]}{dt} = k_{7-}S[1] - (2k_{7+}[ca^{2+}])S[0]$$

$$d\frac{S[1]}{dt} = 2[ca^{2+}]k_{7+}S[0] + 2b_7k_{7-}S[2] - (k_{7+}[ca^{2+}] + k_{7-})s[2]$$

$$d\frac{S[2]}{dt} = [ca^{2+}]k_{7+}S[1] - (2b_7K_{7-})S[2] \tag{6}$$

Here, S[s7] denotes the proportion of release sites with s7 number of Ca²⁺ ions bound to syt7. We assumed that syt7 bound to Ca²⁺ functions as a catalyst for docking, and thus increases both docking and undocking rates. In this way, all proteins included in the model function by lowering an energy barrier (for docking for syt7 and for fusion for Doc2α and syt1/2). From these states, the docking and undocking rates can be computed following

$$docking = k_{docking} \sum_{s7=0}^{2} S[s7] f_7^{s7}$$

$$undocking = k_{undocking} \sum_{s7=0}^{2} S[s7] f_7^{s7} \tag{7}$$

In these equations, $f_7$ is the factor describing how much an individual Ca²⁺ ion bound to syt7 increases the rates of vesicle docking and undocking.

To compute the steady state of the system, all reaction rates (from one state to another, except the Ca²⁺-binding states of syt7) were rewritten into an intensity matrix, Q, in which element i,j describes the rate of transitioning from state i to state j. Subsequently, steady state was computed using the equation

$$\pi_{ss} = \pi_i e^{Qt} \tag{8}$$

With $\pi_i$ a vector of 24 zeros, corresponding to each of the release site states (independent of syt7). The first element of this vector was set to total number of release sites in the model. t was set to 1000s. This large evaluation time was used to ensure that steady state was reached. The resulting $\pi_{ss}$ corresponds to the distribution of the different release site states described in *Equations 1–3* at steady state. The steady state of syt7 was solved separately by setting the syt7 unbound state (S[0]) to 1. Subsequently S[1] and S[2] were computed by

$$S[1] = \frac{2[Ca^{2+}]k_{7+}}{k_{7-}}$$

$$S[2] = \frac{[Ca^{2+}]k_{7+}}{b_7k_{7-}} \tag{9}$$

Next, S[0], S[1], and S[2] were divided by the sum of those three elements to obtain a distribution of the different syt7 bound states at steady state. The steady state of the system was computed for a resting Ca²⁺ concentration of 0.05 µM (*Helmchen et al., 1997*).

With the steady-state calculation as a starting point, the differential equations were solved in MATLAB (2020b) using ode15s (MaxStep = 1e⁻⁵, RelTol = 1e⁻⁵, AbsTol = 1e⁻⁵) and a Ca²⁺ transient corresponding to 10 APs at 20 Hz as input. The Ca²⁺ transient for a single AP was generated using a normal distribution (mean = +1.5 ms, sd = 0.2 ms), which was scaled to obtain the desired amplitude (*Table 1*). To this Ca²⁺ transient, a residual signal was added, which followed an exponential decay:

$$Ca^{2+}_{residual}(t) = 1_{t \geq t0} Ae^{\frac{-(t0+t)}{tau}} \frac{t-t0}{t-t0+km} \tag{10}$$

with, t0 the starting times of the residual $Ca^{2+}$-signal (+1.6 ms from the onset of the AP), tau the decay time constant, t time, and A the amplitude of the residual $Ca^{2+}$ signal. The second term in the equation ($\frac{t-t0}{t-t0+km}$) ensures smoothing of the $Ca^{2+}$ transient. In this term, km was set to 0.1 ms. To the entire $Ca^{2+}$ transient, a basal concentration of 0.05 μM was added (*Helmchen et al., 1997*).

The simulated cumulative number of fused vesicles (F, in *Equations 4*) was interpolated at a sampling rate of $50e^3$ Hz. From this, release rates were computed by multiplying the difference between two time points with the sampling rate. The maximum release rate between stimulus onset (0 ms) and the onset of the next stimulus (+50 ms) was taken as the peak release rate of each stimulus. All release events occurring at +5 ms from stimulus onset to the onset of the next stimulus (+50 ms) were considered as AR events.

Parameters in the model were set to experimentally obtained values, if available. For a list of all model parameters, see *Table 1*. To simplify the model, we assumed that both syt7 and Doc2α have the same $Ca^{2+}$-binding dynamics as the scope of the model was to investigate the apparent role of syt7 and Doc2α in AR when they act on different steps in the reaction scheme. Furthermore, undocking and untethering rate constants were set equal to the docking and tethering rate constants, respectively. This assumption was made to reduce the number of free parameters in the model. The remaining free parameters, $f_2$, $f_7$, $k_{tet}$, $k_{docking}$, amplitude of $Ca^{2+}$ transient, amplitude of residual $Ca^{2+}$ (A), and decay time of residual $Ca^{2+}$ signal (tau), were adjusted manually to obtain close correspondence between model predictions and the phenotypes observed in the experiments presented in this article. More specifically, we decided whether the model could recapitulate experimental observations based on the number of vesicles docking between 5 and 15 ms after an AP (this should be as large as possible for the WT and Doc2α- model, and almost 0 for the syt7-), the peak release rate in response to the first AP (to be equal between all conditions), the ratio between the peak release rate of the 1st and 10th response (depressive phenotype should be the most prominent in syt7- and the least prominent in Doc2α-), and the amount of AR (syt7- and Doc2α- should have approximately half of the total amount of asynchronously released vesicles compared to control). Moreover, the parameter values of the $Ca^{2+}$ transient should be realistic. We do not know the exact values of the $Ca^{2+}$ transient for the hippocampal samples used in this study, but multiple previous studies have provided a range of realistic parameter values (*Brenowitz and Regehr, 2007*; *Helmchen et al., 1997*; *Sabatini and Regehr, 1998*; *Wang et al., 2008*). For simulations of the model lacking Doc2α, $k_{2+}$ was set to 0. To simulate syt7- conditions, f7 was set to 1. The effect of each model parameter on the quantified behavior of the model (number of vesicles docking between 5 ms and 15 ms, total asynchronously released vesicles at the end of the train, peak release rate of the first response, and the ratio between peak release rate of the first and the tenth response) was determined on a range from one-tenth of the selected parameter value to tenfold the selected parameter value (as in *Table 1*).

## Acknowledgements

We thank JD Vevea, MM Bradberry, KC Courtney, D Larson, C Greer, RA Pearce, and MJ Seibert for help with iGluSnFR imaging, maintaining mouse strains, and data analysis and interpretation. We thank MJ Seibert for developing the polymerase chain reaction protocol for Doc2α genotyping. We thank S Myers for recommending the alkaline lysis DNA prep protocol for genotyping. We thank all other members of the Chapman and Watanabe labs for helpful feedback and discussions. The EM ICE high-pressure freezer was purchased partly with funds from an equipment grant from the National Institutes of Health (S10RR026445) awarded to SC Kuo. This article is subject to HHMI's Open Access to Publications policy. HHMI lab heads have previously granted a nonexclusive CC BY 4.0 license to the public and a sublicensable license to HHMI in their research articles. Pursuant to those licenses, the author-accepted manuscript of this article can be made freely available under a CC BY 4.0 license immediately upon publication.

## Additional information

### Funding

| Funder | Grant reference number | Author |
|---|---|---|
| Howard Hughes Medical Institute | Investigator | Edwin R Chapman |
| National Institutes of Health | R01 MH061876 | Edwin R Chapman |
| National Institutes of Health | R35 NS097362 | Edwin R Chapman |
| National Institute of Neurological Disorders and Stroke | NS105810 | Shigeki Watanabe |
| National Institute of Neurological Disorders and Stroke | NS111133-01 | Shigeki Watanabe |
| Alfred P. Sloan Foundation | Fellow | Shigeki Watanabe |
| McKnight Endowment Fund for Neuroscience | Scholar | Shigeki Watanabe |
| Novo Nordisk Fonden | Young Investigator Award NNF19OC0056047 | Alexander M Walter |
| Deutsche Forschungsgemeinschaft | Emmy Noether Program 261020751 | Alexander M Walter |
| Deutsche Forschungsgemeinschaft | TRR 186 278001972 | Alexander M Walter |
| Esther A. and Joseph Klingenstein Fund | Fellow | Shigeki Watanabe |
| National Institutes of Health | T32 GM007445 | Grant F Kusick |
| National Science Foundation | Graduate Research Fellowship Program 2016217537 | Grant F Kusick |
| National Science Foundation | 1727260 | Shigeki Watanabe |
| National Institutes of Health | S10RR026445 | Shigeki Watanabe |
| National Institute of Neurological Disorders and Stroke | NS132153-01 | Shigeki Watanabe |

The funders had no role in study design, data collection and interpretation, or the decision to submit the work for publication.

### Author contributions

Zhenyong Wu, Formal analysis, Investigation, Visualization, Writing – original draft, Writing – review and editing, Performed the iGluSnFR and electrophysiology experiments, analyzed the corresponding data, and generated the corresponding figures; Grant F Kusick, Conceptualization, Formal analysis, Investigation, Visualization, Writing – original draft, Writing – review and editing, Performed the electron microscopy experiments, with help from Sumana Raychaudhuri, analyzed the corresponding data with Shigeki Watanabe, and generated the corresponding figures with Shigeki Watanabe. Conceived of of the sequential action of Doc2a and syt7; Manon MM Berns, Software, Formal analysis, Investigation, Visualization, Writing – original draft, Writing – review and editing, Developed and coded the mathematical model, performed the simulations, and generated the corresponding figures; Sumana Raychaudhuri, Resources, Investigation, Performed neuronal cell culture for and assisted with the electron microscopy experiments; Kie Itoh, Resources, Performed neuronal cell

culture for the electron microscopy experiments; Alexander M Walter, Supervision, Funding acquisition, Writing – original draft, Project administration, Writing – review and editing, Supervised the simulations; Edwin R Chapman, Conceptualization, Supervision, Funding acquisition, Writing – original draft, Project administration, Writing – review and editing, Supervised the iGluSnFR and electrophysiology experiments. Conceived of the original investigation comparing Doc2a and syt7; Shigeki Watanabe, Conceptualization, Formal analysis, Supervision, Funding acquisition, Visualization, Writing – original draft, Project administration, Writing – review and editing, Supervised the electron microscopy experiments and participated in analyzing the corresponding data and generating figures

### Author ORCIDs
Grant F Kusick ⓘ http://orcid.org/0000-0002-4312-3495
Manon MM Berns ⓘ https://orcid.org/0000-0003-2998-4202
Kie Itoh ⓘ http://orcid.org/0000-0003-4379-400X
Alexander M Walter ⓘ https://orcid.org/0000-0001-5646-4750
Edwin R Chapman ⓘ https://orcid.org/0000-0001-9787-8140
Shigeki Watanabe ⓘ http://orcid.org/0000-0001-7580-8141

### Ethics
All animal care was performed according to the National Institutes of Health guidelines for animal research with approval from the Animal Care and Use Committees at the Johns Hopkins University School of Medicine and the University of Wisconsin, Madison (Assurance Number: A3368-01).

Reviewer #1 (Public Review): https://doi.org/10.7554/eLife.90632.3.sa1
Reviewer #2 (Public Review): https://doi.org/10.7554/eLife.90632.3.sa2
Author Response https://doi.org/10.7554/eLife.90632.3.sa3

## Additional files

### Supplementary files
• MDAR checklist

### Data availability
All data generated during this study are included in the manuscript and supporting files; full data tables underlying all figures are in the source data. Custom R, Matlab and Fiji scripts for electron microscopy analysis are available on GitHub (copy archived at *Watanabe, 2024*). More details can be found in *Watanabe et al., 2020*. All code for the mathematical model is available in *Figure 7—source code 1*.

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
