## [Editor Report · eLife assessment]

In this **important** study, the authors identify distinct roles for the calcium sensors synaptotagmin 7 and Doc2α in the regulation of asynchronous release and calcium-dependent synaptic vesicle docking in hippocampal neurons. The current work adds to the field by placing the role of the two proteins in a new context, where synaptotagmin 7 acts to promote synaptic vesicle docking and capture after a stimulus, while Doc2α has a role in specifically driving the asynchronous component of release as a calcium sensor. The methods, data, and analyses provide **convincing** support for the conclusions.

---

## [Referee Report · Reviewer #1 (Public Review)]

Summary:

In the current manuscript, the authors find distinct roles for the calcium sensors Syt7 and Doc2alpha in the regulation of asynchronous release and calcium-dependent synaptic vesicle docking in hippocampal neurons. The authors data indicate that Doc2 functions in activating a component of asynchronous release beginning with the initial stimulus, while Syt7 does not appear to have a role at this early stage. A role for Syt7 in supporting both synchronous and asynchronous release appears during stimulation trains, where Syt7 is proposed to promote synaptic vesicle docking or capture during stimulation. Doc2 mutants show facilitation initially during a train and display higher levels of synchronous release initially, before reaching a similar plateau to controls later in the train. The authors contribute the increased synchronous release in Doc2 mutants to Syt1 having access to more SVs that can fuse synchronously. In contrast, Syt7 mutants show depression during a train, and continue to decline during stimulation. The authors contribute this to a role for Syt7 in promoting calcium-dependent SV docking and capture that feeds SVs to both synchronous and asynchronous fusion pathways. Importantly, phenotypes of a double Doc2/Syt7 mutant collapse onto the Doc2 phenotype, suggesting the two proteins are not additive in their role in supporting distinct aspects of SV release. Rapid freeze EM after stimulation provides support for a role for Syt7 in SV docking/capture at release sites, as they display less docked SVs after stimulation. In the case of Doc2, EM reveals fewer SVs fusion pits later during a stimulation, consistent with fewer asynchronous fusion events. The authors also provide modeling that supports aspects of their conclusions from the experimental data. I cannot evaluate the modeling data or the specific experimental subtlities of the GluSnFR quantification approach, as these are outside of my reviewer expertise.

Strengths:

The use of multiple approaches (optical imaging, physiology, rapid freeze EM, modeling, double mutant analysis) provides compelling support for distinct roles of the two proteins in regulating SV release.

Weaknesses:

Some of the phenotypes for both Doc2 and Syt7 mutants have been reported in the authors' prior publications. It is not clear how well the GluSnFR approach is for accurately separating synchronous versus asynchronous release kinetics. The authors also tend to overstate the significance of the two proteins for asynchronous release in general, as a significant fraction of this release component is still intact in the double mutant, indicating these two proteins are only part of the asynchronous release mechanism.

---

## [Referee Report · Reviewer #2 (Public Review)]

Summary:

The goal of this study is to provide a deeper understanding of the roles of syt7 and Doc2 in synaptic vesicle fusion. Depending on the system studied, and the nature of the preparation, it appears that syt7 functions as a sensor for asynchronous release, synaptic facilitation, both processes, or neither. The perspective offered by Chapman, Watanabe, and colleagues varies from those previously published, and is therefore novel and interesting.

Strengths:

The strengths of the study include the complementary imaging and electrophysiology approaches for assessing the function of syt7, and the use of appropriate knockout lines. High resolution imaging approaches to measure synaptic activity is also a strength.

Weaknesses:

It is not clear to this reviewer that the computational modeling effort is important or even necessary. The study also attempts to derive kinetic information (on the ms time scale) from EM. While the interpretations are not unreasonable, they should be taken with some caution.

Overall, the study does a good job of attempting to resolve the various ambiguities existing in the field regarding the potential roles of syt7 and Doc2 in membrane fusion. There are, of course, a great number of proteins which have been identified to act at fusion sites to drive or otherwise modify release phenotypes. Efforts such as this are going to become increasingly important as we work to attribute discrete roles to each one.

---

## [Author Response]

The following is the authors’ response to the original reviews.

First, we discovered several erroneous duplicate values in our source data sets from figures S1, 2, 4, and 8, due to mistakes from MATLAB analysis. We have re-analyzed the data and corrected these errors; since limited values in each data set changed, the results were unaffected. The changes are reflected in updated figures and source data.

Overall, the reviewers gave a positive assessment of our work, but had reservations about:

(1) Specifics of the iGluSnFR data and analysis

(2) Overstatement/oversimplification of the importance of syt7 and Doc2

(3)The strength and interpretation of the EM data (4) The relevance and parametrization of the modeling data

(1) We have clarified aspects of the iGluSnFR data and analysis in the point-by-point response, as well as in the manuscript.

(2) We have toned down our statements about the role of syt7 and Doc2 throughout, and emphasized that the DKO data are conclusive and reveal that there must be additional Ca2+ sensors for AR. We have also added to the discussion, noting syt3 as a strong candidate to perform a function analogous to syt7 (to regulate docking), along with another protein (or proteins) performing a role similar to Doc2 (directly in fusion) that has not been identified as a candidate in the field yet.

(3) We feel the EM data are consistent with the model as much as they could be, and while a sequence of events can only be inferred from time-resolved EM, we believe our work falls in the scope of reasonable interpretation. However, upon reexamining the terminology of ‘feeding’ and related discussion, we realized this could be misleading, so these sections have been revised.

(4) We have improved the description and interpretation of the model in the manuscript and provide a detailed rationale of our approach in the point-by-point-response.

**Reviewer #1 (Recommendations For The Authors):**
Major points:(1) It is surprising the optical GluSnFR approach reports so much asynchronous release in control hippocampal neurons after single stimuli (36% of release). This seems much higher than what is observed at most synapses, where asynchronous release is usually less than 5% of the initial response to the first evoked stimuli. Any thoughts on why the GluSnFR approach reports such a high level of asynchronous release? Could the optical approach be slower in activation kinetics in some cases, which artificially elevates the asynchronous aspect of fusion? This seems to be the case, given electrophysiology recordings in Figure 3 show the asynchronous release component as ~10% in controls at the 1st stimuli (panel C).

The reported proportion of asynchronous release from cultured hippocampal neurons varies, contingent upon a range of factors (calcium concentration, how asynchronous release is quantified, etc). However, we would argue that there is considerable evidence for a higher percentage of asynchronous release (more than the <5% indicated by the referee) at synapses in the hippocampus. In our previous work on Doc2 using electrophysiology in cultured hippocampal neurons (Yao et al., 2011, Cell), it was noted that there is an approximate 25% incidence of asynchronous release after a single action potential. Furthermore, Hagler and Goda also reported a 26% ratio of asynchronous neurotransmitter release, also from cultured hippocampal neurons (Hagler and Goda, 2001, J Neurophysiol.).

We also point out that another study using iGluSnFR to measure synchronous/asynchronous release ratios, with more sophisticated stimulation, imaging, and analysis procedures than ours, found an average ratio of synchronous to asynchronous release that is in-line with our values, with considerable variability among individual boutons (Mendonça et al., 2022; 25% asynchronous release after a single action potential). We feel that iGluSnFR is actually the superior approach (barring specialized e-phys preparations that can measure quantal events at individual small synapses; please see Miki et al., 2018), as it directly measures the timing of individual release events at individual boutons. By comparison, in most electrophysiology experiments there is a large peak of synchronous release from many synapses. iGluSnFR also bypasses postsynaptic considerations such as receptor kinetics and desensitization, or asynchronous release being poorly aligned to AMPA receptors, per a recent study of ours (Li et al., 2021), and a study showing 25% of asynchronous release occurs outside the active zone (Malagon et al., 2023). All these factors could obscure asynchronous release or otherwise make it difficult to measure by electrophysiology. To our knowledge, the approach in Miki et al., 2018 best bypasses these limitations, though the data in that study are from exceptionally fast and synchronous cerebellar synapses, and so cannot be directly compared to our findings. Thus, it is possible that iGluSnFR can report more asynchronous release than electrophysiological recordings, but this may actually reflect real biology.

This being said, after considering the reviewer’s points we realized that our analysis method likely underestimates the total amount of synchronous release when using the high-affinity sensor (Figure 1). We quantify release by ‘events’ (that is, peaks), which does not take into account multiquantal peaks resulting from near-simultaneous multivesicular release. We have previously determined by quantal analysis that most synchronous peaks after a single action potential are multiquantal, while for asynchronous release there are still multiquantal events but they are in the minority (Vevea et al., 2021; Mendonça et al., 2022). So, in our data sets, the total amount of synchronous release is underestimated more so than asynchronous release. Thus, 37% asynchronous release is probably an overestimate, which explains the 12% difference compared to Mendonça et al., 2022, who used sophisticated quantal analysis (though that study also was performed at room temperature, which could also cause differences). We have now pointed this out in the text:

“This ratio of synchronous to asynchronous release is likely an underestimate, since our analysis only counts the number of peaks (‘events’) and does not take into account multiquantal peaks resulting from near-simultaneous multivesicular release. We have previously determined by quantal analysis that most synchronous peaks are multiquantal after a single action potential, while for AR there are still multiquantal events but they are in the minority (Vevea et al., 2021). So, in our measurements, the total amount of synchronous release is underestimated; sophisticated quantal analysis using the A184V iGlusnFR recently found the percentage of total release that is AR to be ~25%, with otherwise similar results to ours (Mendonça et al., 2022) . Nonetheless, this approach faithfully distinguishes synchronous from asynchronous release…”

However, while this method underestimates total synchronous release, it does not misclassify synchronous events as asynchronous because of kinetics. Even the slower iGluSnFR variant does not have a rise time that would misrepresent a synchronous event as asynchronous (Marvin et al., 2018). Mendonça et al (2022) note that averaged iGluSnFR traces for the A184V are biphasic, with the transition from fast to slow component occurring around 10 ms. These authors also determined that the temporal resolution of glutamate imaging is actually limited by the frame rate, not the biosensor, and based on simulations found that detection time was biased in their data to be about 1 ms earlier than the actual timing of release events.

The reviewer’s final point about Figure 3 is a misunderstanding, as these are data from iGluSnFR, not electrophysiology. The asynchronous proportion in these experiments is ~10% because, as noted in the manuscript, we used a faster, lower-affinity variant of iGluSnFR in train stimulation experiments (Figure 2). In contrast to the high-affinity sensor, as explained above, in our analysis this variant would be expected to underestimate the amount of asynchronous release because it fails to detect many uniquantal release events (presumably those further from the focal plane, with too little fluorescence to reach our detection threshold) as evidenced by the fact that the apparent mini rate is much lower as measured by this sensor compared to higher-affinity variants. Since synchronous peaks are mostly multiquantal after a single action potential, while asynchronous peaks are mostly uniquantal, a fraction of release going undetected results in mostly smaller synchronous peaks, which are counted the same in our analysis while many asynchronous peaks are missed entirely. We have added a bit more clarification in the text to avoid confusion on this point:

“This sensor underestimates the fraction of AR (~10% of total release for a single action potential) as compared to the A184V variant used above that overestimates the fraction of AR (~35% of total release for a single action potential). This is because it is less sensitive and misses many uniquantal events; as discussed above, our analysis quantifies release by number of peaks, and most synchronous peaks are multiquantal after a single action potential, while most AR peaks are uniquantal (Vevea et al., 2021). Still, the S72A variant reported the same phenotypes as the A184V variant after the first action potential (Fig. 3B, C).”

As discussed above, we think the synchronous-to-asynchronous ratio is actually harder to determine with electrophysiology, and the preparations are different (acute slice vs dissociated culture); still, our electrophysiological measurements are in line with the iGluSnFR data: 29% for Figure 2 and 26% from the first action potential of Figure 4. These values also agree with the findings from Yao et al. (2011) and Hagler and Goda (2001), discussed above.

Finally, the ultimate goal of our study was to measure the effects of deleting Doc2 and syt7 on synchronous and asynchronous release, not to measure the exact ratio between the two. If iGluSnFR greatly misreported synchronous events as asynchronous, we would expect the results from the knockouts to diverge between our imaging and electrophysiology data, which they do not. We have also previously applied this approach to syt1 knockouts, showing the characteristic desynchronization of release (Vevea et al., 2020). Furthermore, the high-affinity and low-affinity iGluSnFR variants, which as discussed above in our analysis overestimate and underestimate the fraction of release that is asynchronous, respectively, both reported the same phenotypes.

(2) In the acute hippocampal physiology traces, it looks like the effect on cumulative release in Doc2A mutants only appears around ~40 msec after stimulation. This is a relatively late phase of asynchronous release. Any reason this effect does not show up sooner, where most asynchronous fusion events occur, or is this due to some technical aspects of the physiology clamp that masks earlier components?

The reviewer is correct, although the curves actually diverge at around 30 ms (see image below). This can be attributed to the fact that the EPSCs in our recordings are broad, probably because of the large number of different synaptic inputs captured in our stimulation and recording paradigm (note that the currents are also quite large), resulting in a broad spread in the timing of release. That is to say, synchronous release is likely still occurring fairly late into the trace, obscuring any changes in asynchronous release earlier than 30 ms. This is not related to Doc2 specifically, as the EGTA charge transfer curve also diverges from the control curve at the same time. This EGTA control gives us confidence that our broad EPSCs still faithfully report synchronous and asynchronous release, even if the exact timing is spread-out to some extent.

(3) How do the authors treat multi-vesicular release in their synchronous/asynchronous quantification? It was not clear from the methods section. Many of the optical traces show dual peaks - are those that occur in the 10 ms bin assigned to synchronous and those outside to asynchronous? Are the authors measuring the area of the response or just the peak amplitude for the measurements? The methods seem to indicate peak amplitude, but asynchronous is better quantified with area measurements for electrophysiology.

This is an excellent point by the reviewer, and in the Methods we now explicitly state how we treat multivesicular release/multiple peaks in our analysis. Release timing is assigned based on peak timing, including when there are multiple peaks at the same bouton.

“Timing of release was determined based on the frame in which the signal peaked, including for dual peaks in the case of synchronous and asynchronous release at the same bouton.”

Regarding the comparison to area measurements for electrophysiology, we agree with the reviewer, which is why we used such an approach for our electrophysiological data. However, a key advantage of iGluSnFR is the ability to resolve individual quantal events (or, as is often the case for synchronous release, simultaneous multiquantal events), so temporal binning of the peaks is the appropriate analysis approach regarding these data. This is comparable to the analysis used for electrophysiology recordings of responses from single small synapses, which also detects individual quantal events, where release timing is calculated as the latency between the stimulus and the beginning of each EPSC (Miki et al., 2018).

This leaves the general concern that multiple vesicle fusions at the same bouton that occur milliseconds apart could blur together and make it more difficult to accurately determine release timing, particularly with the slower sensor used in the single-stim experiments in Figure 1. We believe this is not a major concern, since we also performed experiments with the much faster sensor, S72A which can resolve peaks from 100 Hz stimulation (Marvin et al., 2018). Furthermore, while the peak-calling method we used is crude by comparison, the synchronous/asynchronous ratio we report is similar to that of Mendonça et al. (2022) who used a higher frame rate and deconvolution to produce more easily distinguishable quanta when synchronous and asynchronous release occur at the same bouton after the same action potential.

(4) It would be relevant to show that calcium binding mutations in Syt7 do not support SV docking/capture in the current assays, given some evidence for Syt7 calcium-independent activities has been reported in the field.

To our knowledge, when using the correct mutations to block calcium binding, none of the reported syt7 knockout phenotypes (including those reported by our laboratory in Liu et al., 2014) have ever been rescued. However, this does not formally rule out a calciumindependent role in transient docking. For the EM data, we originally considered including rescue experiments with normal and non-calcium binding mutants of both syt7 and Doc2 in our study. However, our EM approach is spectacularly expensive and labor-intensive and such experiments would as much as triple the amount of EM work in the study. We plan on doing such experiments, and there is a great deal of additional structure-function work to be done on both these proteins. We feel that reassessing the calcium binding mutants with iGluSnFR and zap-andfreeze falls into the scope of this future work. For now, this as a limitation of the current study.

(5) The authors are not consistent in how they describe the role of the two proteins in asynchronous release, with the reader often drawing the impression that these two proteins solely mediate this aspect of SV fusion. As the authors note, some synapses do not require Syt7 or Doc2 for SV release, indicating different asynchronous sensors or molecular components at distinct brain synapses. Indeed, asynchronous release is only reduced, not eliminated, in the double mutants the authors report, so other components are at play even in these hippocampal synapses. The authors should be more consistent in noting this in their text, as the wording can be confusing as noted below:

"Together, these data further indicated that AR after single action potentials is driven by Doc2α, but not syt7, in excitatory mouse hippocampal synapses."

"after a single action potential, Doc2α accounts for 54-67% of AR at hippocampal excitatory synapses, whereas deleting syt7 has no effect."

"This, along with our finding that syt7/Doc2a DKOs still had remaining AR, raises the possibility that there are other unidentified calcium sensors for AR."

We have made adjustments throughout to not overstate the role of syt7 and Doc2, including at the locations the reviewer points out. This is an important point from the reviewer, and not just to avoid misleading readers. It is itself interesting; in the original manuscript we should have emphasized, far more than we did, that the DKO experiments strongly point to asyet-unidentified proteins being involved in asynchronous release. This has been rectified in the revised text: we now emphasize that another calcium sensor for asynchronous release is likely present at all relevant points in the manuscript.

(6) Given the authors' data, I don't think it's fair to say "raises the possibility" of other AR sensors, as almost 50% of AR remained in the Doc2A mutant in some of the experimental approaches. Clearly, other AR calcium sensors or molecular components are required, so better to just state that in the 1st paragraph of the discussion with something like:"Given syt7/Doc2a DKOs still had remaining AR, further work should explore the diversity of synaptic Ca2+ sensors and how they contribute to heterogeneity in synaptic transmission throughout the brain."

We agree; this was poor phrasing on our part. We meant to imply that there *may* be proteins that have not even been considered, because it is also technically possible that the remaining asynchronous release is supported by the known machinery (i.e., syt1). We have changed “raises the possibility” to “indicates”.

Minor points:(1) Remove "on" from the abstract sentence "Consequently, both synchronous and asynchronous release depress from the second pulse on during repetitive activity".

We have changed “on” to “onward” to reduce ambiguity.

(2) Shouldn't syt7 be Syt7 and syt1 be Syt1 when referring to the proteins?

To our knowledge there is not a hard-and-fast convention for non-acronym mouse protein abbreviations. The technically correct full name is lowercase, so we find it reasonable to use lowercase for the abbreviation.

(3) Both calcium and Ca2+ are used in the manuscript - better to stick to one term throughout.

We thank the referee for catching this error; we now use only “Ca2+” throughout our study.

**Reviewer #2 (Recommendations For The Authors):**
(1) While the GluSnFR experiments appear to be well done, what is striking is the relatively small and "jagged" fluorescent responses. Are the authors concerned that they are missing many fast (with peaks occurring within 10 ms) synchronous events and incorrectly identifying them asynchronous? If this is not a concern, why not?

With respect to the small raw responses, this is the nature of measuring individual quanta from individual boutons while imaging at 100 Hz, even with the excellent signal-to-noise ratio of the iGluSnFR variants we used.

As far as kinetics, as noted in the response to Reviewer 1 point #1, even the slower iGluSnFR variant has a rise time fast enough that it cannot misrepresent a synchronous event as asynchronous (Marvin et al., 2018). This threshold for iGluSnFR has been used by others: see Mendonça et al., 2022, who note that averaged iGluSnFR traces are biphasic, with the transition from fast to slow component occurring around 10 ms. The ‘jaggedness’ is in large part due to the frame rate (100 Hz); Mendonça et al., 2022 used 250 Hz and deconvolution to produce smoother, cleaner traces, but still achieved similar results to us.

Finally, we reiterate what we wrote in response to Reviewer 1 point #1: “the ultimate goal of our study was to measure the effects of deleting Doc2 and syt7 on synchronous and asynchronous release, not to measure the exact ratio between the two. If iGluSnFR misreported synchronous events as asynchronous, we would expect the results from the knockouts to diverge between those data and our electrophysiology data, which they do not. We have also previously applied this approach to syt1 knockouts, showing the characteristic desynchronization of release (Vevea et al., 2020). Also, the phenotypes reported by the faster and slower iGluSnFR variants were identical. ”

(2) On page 6, I'm not sure I would agree that short-term plasticity is "so catastrophically disrupted". It is probably enough to say that plasticity is disrupted in the ko.

We argue that syt7 knockout causes the most severe phenotype specific to short-term plasticity so far described (that is, without affecting initial release probability), but we have changed “catastrophically” to “strongly”.

(3) Differences in the post-stim number of "docked" vesicles between conditions are, in absolute numbers, very small. For example, it seems that the number of docked vesicles goes from ~ 2.2 prior to stimulation, to ~ 1.5 in the first 5 ms window following stimulation. While this number may be statistically significant, I worry about bias and sampling errors. It is comforting that images are randomized prior to analysis. Nevertheless, the differences are very small and this should be explicitly acknowledged.

This ~40% decrease in number of docked vesicles in dissociated cultured hippocampal neurons has been consistent throughout all our studies using flash-and-freeze and zap-and-freeze electron microscopy (Watanabe et al., 2013; Kusick et al., 2020, Li et al., 2021), as well as those of other labs (Chang et al., 2018). Statistically, 40% is far beyond the limit to detect differences between samples with 200-300 synapses quantified per condition and an average of ~2 docked vesicles per image. The low absolute number of docked vesicles per synaptic profile (since the 40 nm section only captures a portion of the active zone, which contain an average of 12 docked vesicles in total; Kusick et al., 2020) is not relevant except that it does reduce the statistical power to detect differences, but this is compensated for by the huge number of images we capture and annotate per sample. We are able to detect differences in fusion and endocytic pits (albeit with much less precision and sensitivity), such as the Doc2 phenotype in this study, even though these events are an order of magnitude rarer than docked vesicles. Biologically, in our view, a 40% reduction in all docked vesicles across all synapses, considering that the majority of synapses do not have even 1 vesicle fusion, after only a single action potential, is substantial. We have even been puzzled why there is such a large decrease, but as stated above this result has been consistent for a decade of using this approach. For comparison to the magnitude of baseline docking changes in mutants, this 40% is similar to the effect of deleting synaptotagmin 1 (Imig et al, 2014; Chang et al, 2018; note in Imig et al., considered a gold standard in the field, the average number of docked vesicles per tomogram is ~10, but there are fewer than 25 tomograms per sample, so the actual amount of sampling in our data set is slightly greater).

(4) The related point is that how can one know about the "transient" nature of vesicle docking when the analysis is performed on completely different sections from different cells? Moreover, what does it mean that the docked granules have recovered or not recovered (abstract)? This should be explained in more detail.

This is a fundamental difficulty of interpreting time-resolved electron microscopy data. We cannot observe a sequence of events at any given synapse, but only try to measure each time point as accurately as we can and interpret the data.

By ‘recovery’ we simply mean that the number of docked vesicles at a given time point after stimulation is similar to the no-stimulation baseline. We have replaced ‘recovery’ in the abstract with ‘replenishment’ to avoid confusion.

We now realize that in the context of this study the term ‘transient docking’ is confusing, since we only measured out to 14 ms in this study. In experiments with samples frozen at 5 ms, 14 ms , 100 ms, 1,s and 10 s, the return to baseline at 14 ms appears temporary, since samples frozen at 100 ms have a similar reduction of docked vesicles as those at 5 ms (Kusick et al., 2020). The number of vesicles again returns to baseline at 10 s, so we used the term ‘transient docking’ to distinguish the recovery at 14 ms from the slower and presumably permanent return to baseline that takes 10 s. The apparently temporary nature of this process is why we believe it contributes to facilitation, which likewise peaks soon after stimulation and decays over the course of ~100 ms.

To make the transient docking terminology less confusing, we have removed the word‘transiently’ from the title and added a clarification of what transient docking is when it is first mentioned:

“vesicles can dock within 15 ms of an action potential to replenish vacated release sites and undock over the next 100 ms”

As noted by the reviewer, such a sequence of events, where vesicles dock within 14 ms, then undock over the course of 100 ms, then dock again over the course of 10 s, is an inference, but is based on predictions from electrophysiological data and modeling (see Silva, Tran, and Marty, 2021 for review; those authors use the term ‘calcium-dependent docking’ but this refers to the same process), and as yet there is no way to directly observe vesicle dynamics at synapses down to nanometer resolution in live cells.

On the reviewers recommendation we have removed references to syt7 ‘feeding’ vesicles from the abstract and the beginning of the “physiological relevance” section of the discussion. This phrasing could imply a direct molecular pipeline between syt7 and syt1/Doc2, which is a misrepresentation of our actual model that syt7 simply helps recruit docked vesicles.

“These findings result in a new model whereby syt7 drives activity-dependent docking, thus providing synaptic vesicles for synchronous (syt1) and asynchronous (Doc2 and other unidentified sensors) release during ongoing transmission.”

“In the case of paired-pulse facilitation it can supply docked vesicles for syt1-mediated synchronous release to enhance signaling; it likely functions in the same manner to reduce synaptic depression during train stimulation. In the case of AR, syt-7-mediated docked vesicles can be used by Doc2α, which then directly triggers this slow mode of transmission.”

(5) In this study, docking is phenomenologically defined and, therefore, arbitrary; vesicles are defined as docked if there is no space between them and the plasma membrane. What happens if the definition is broadened to include some small distance between the respective membranes? Does the timecourse of "recovery" change?

We always quantify at least all vesicles within 100 nm of the active zone; these data are shown in Figure S6D. We show only docking in the main figures because, consistent with our previous work and as stated in the text, we found no change in the number of vesicles at any distance from the plasma membrane at the active zone after stimulation, nor did we find any difference in the mutants. In our previous work on syt7 (Vevea et al., 2021) we quantified all the vesicles within the synapse and also found no differences after stimulation or in the KO further from the active zone.

The reviewer is correct that the term ‘docking’ at synapses is often used quite arbitrarily; even among morphological studies the definition is inconsistent. We consider our strict docking definition that we explain in the manuscript (in high-pressure-frozen and freeze-substituted samples) of no visible distance between membranes to be less arbitrary, since only the number of these attached vesicles decreases after stimulation (Watanabe et al., 2013, Kusick et al., 2020, Li et al., 2021, this study) and in SNARE knockouts (Imig et al., 2014). Broadening the definition, as is done in some other studies (for example Chang et al., 2018), retains the effect, since the majority of vesicles within 10 nm are at ~0 nm, but again all that is actually changing is the number of vesicles at ~0 nm.

(6) My overall impression is that this model is not adding much to the story. Specifically, the model was not fit to any data and has a huge number of states and free parameters given the dynamics that it is trying to capture (ie I think this is overkill). Many of the free parameters were arbitrarily constrained with little to no justification and there was minimal parameter space exploration, in part because the model wasn't being quantitatively constrained to any data. While advertised to be a 3-state model, there is a combinatorial explosion of substates by distinguishing between levels of calcium occupancy simultaneously in three separate calcium sensors so that one ends up with 9 empty states, 9 tethered states, and 45 docked states for a total of 63 distinguishable states. At 63 states and 21 free parameters, one could of course model just about any dynamics imaginable. But the relatively simple dynamics of AR and its perturbation by removal of Doc2 and Syt7 can likely be captured with far fewer states and parameters (such as Neher's recent proposal). Specifically, starting with the Neher ES-LS-TS model along with adding a transient labile docked state affected by Syt7 and Doc2 (TSL in Neher nomenclature), I wonder if the authors could more or less capture what they are observing during stimulus trains. The advantage of a minimal model is that readers don't have to struggle with fairly elaborate systems of differential equations and parameter plots to get a feel for what's going on. Especially since the point of this model is to develop intuition rather than to capture with physical accuracy exactly what is transpiring at a docked vesicle (which would require many more details excluded from the current model).

We would like to thank the reviewer for pointing out unclarities and mistakes in the description of the model. We have worked on improving on these points. We now more elaborately explain why we have made certain assumptions and what decisions we have made to constrain the parameter values in the model. As the reviewer points out other models might also work in explaining the dynamics of the experimental data presented in this paper. Thus, we agree that it is unlikely that this theory and model implementation is the only one that can account for the observations. With this model we aimed to investigate whether the theory proposed based on the experimental data could indeed reproduce the dynamics that are observed experimentally. In the section below we will briefly explain why we made different decisions in constructing the model to comment on the reviewer’s concerns. We will also discuss more precisely what adjustments we have made to the model’s description to improve its readability and be open about its limitations.

One of the main concerns of the reviewer is that the model has many states and free parameters, some of which are poorly constrained. We agree that the model indeed contains many states. However, in essence, the model corresponds to a two-step docking model, in which SVs get tethered to an empty release site and subsequently dock/prime in a fusion-competent state. This structure of the model corresponds to the ES-LS-TS model (Neher and Brose 2018, Neuron) mentioned by the reviewer or the replacement-docking model (Miki et al., 2016, Neuron). As the reviewer points out, by making the transition rates calcium-dependent in those models, we would indeed be able to capture similar dynamics with these models as with ours. However, instead of directly implementing calcium-dependent rates, we let the rates depend on the number of calcium ions bound to syt7, Doc2 and Syt1. We decided to do so, as some information on the calcium binding dynamics of these proteins is available. By simulating the calcium binding to the proteins explicitly we could integrate this knowledge into our model. Moreover, by explicitly simulating calcium-binding to these proteins, we included the time it takes before a new steady state-binding occupancy is reached after a change of calcium levels. Especially for Ca2+ sensors with slow kinetics such as, syt7 and Doc2, this is crucial. These properties are highly relevant for asynchronous release (which we quantified as the release >5 ms after onset of AP). The consequence is that because of combinatorics (e.g., if we assume 5 calcium ions to bind to syt1 and 2 to Doc2 this leads to 24 different states), explicit simulation of all relevant states extends the number of potential different states a vesicle can be in. In the main text of the manuscript, we added this explanation on why we decided on the structure of the model as it is presented and discussed it in context of other previous models.

Our decision to simulate calcium binding to syt1, syt7 and Doc2 also increased the number of parameters in our model. As the reviewer points out, the large number of parameters in our model compared to the relative low number of features in the experimental behavior the model is compared to – is a limitation. However, after thorough exploration of the model, we are certain that the model cannot create any type of desired dynamics. The large number of parameters does make it possible that different combinations of parameter values would lead to similar responses, as can be seen in the parameter space exploration in Figure S9. This means that our modelling effort does not provide estimates of parameter values. We now mention this explicitly in the discussion section of the model. Some of the parameter values we were able to constrain based on previous literature (10 parameters), others were more arbitrary set (8 parameters), and some of them were adjusted to match the experimental data closely (7 parameters). We indicated more clearly now in Supplementary Table 3 to which category each parameter value belongs in table. We determined the values of the model parameters through a manual exploration of the parameter space. One of the main reasons why we decided not to perform a fitting of the model to data obtained in this work is that the obtained parameters would not be informative (e.g., multiple combinations of parameters will lead to similar results). We agree with the reviewer that a direct quantitative comparison between model predictions and experimental data obtained by fitting would be nice. However, fitting the model to experimental data would be close to impossible computationally. This is in part because of the large number of states, but mainly due to the large number of APs that need to be simulated. Especially since the transients in our model have slow and fast parts (the decay of the residual Ca2+-transient, and the peak of the local Ca2+transient), the model is challenging to solve with ODE solvers available in Matlab, even when using a high-performance computer system optimized for parallel computation (32 cores). Moreover, fitting the model to experimental data would require the addition of extra assumptions and parameters to the model. As the experiments are performed using different samples, different parameter settings are probably required (e.g. it is likely that the number of release site or the fusion probability differs between cultured hippocampal neurons and hippocampal slices). Additionally, if we decide to fit the model, we would need to define a cost function (i.e., a quantitative measure of how well the model is fitting to experimental data), which requires us to determine the different weights the different experiments we are comparing our model predictions to have. The decision on how to weight the different types of data is very difficult (not to say arbitrary).

Therefore, we constrained the parameter values in our model based on a manual (but systematic) exploration of the parameter space. The simulations of the model were evaluated based on the increase in the number of docked vesicles between 5 and 15 ms after AP stimulation (this should be as large as possible for the control and Doc2- model, and close to 0 for the syt7- model simulations), the peak release rates in response to the first AP (to be equal between all conditions), the ratio between the peak release rate of the 1st and 10th response (depressive phenotype should be more prominent in the syt7- model simulation and the least in the Doc2- simulation), and the amount of asynchronous release (syt7- and Doc2- simulations should have approximately half of the total amount of asynchronously released vesicles compared to the control simulations). Moreover, the parameter values for the calcium transient should be realistic. We do not know the exact parameter values of the calcium transient in the samples used in the experiments performed here, but previous studies have provided a range of realistic parameter values (Brenowitz and Regehr 2007, PMID: 17652580; Helmchen et al., 1998, PMID: 9138591; Sabatini and Regehr 1998, PMID: 9512051; Wang et al., 2008, PMID: 19118179). Furthermore, we decided to set the parameters describing calcium binding to syt7 and Doc2 to the same values, as the scope of the model was to investigate the role of syt7 and Doc2 in asynchronous release when they act on different steps in the reaction scheme. By using the same parameter values both proteins are identical except for their mechanism of action. We added this section to the methods of the manuscript.

In the parameter space evaluation, we decided to vary parameters one-by-one or in pairs of two. We decided not to further extend the parameter space evaluation as it will be challenging to give a proper interpretation of these results, to visualize them, and to simulate it (computationally expensive).

(7) The graphics, equations, and nomenclature all need some work. The equations aren't numbered or indexed, so I can't really refer to any of them in particular, but the symbols being used generally were not defined well enough for a naïve reader to follow. The 15 diffEQs compressed into a single expression at the bottom of page 19 are basically impenetrable. The 'equation' near the bottom of p. 20 is not an equation - it is a set of four symbols lacking a definition. The fusion rate equation (with f1 and f2 factors) isn't spelled out clearly enough (top of p. 20). Can fusion occur from any of the 45 docked states but just with a different probability? Or does fusion only occur from the 3 states where Doc2+Syt1 Ca occupancy = 5? The graphical representation of Syt7 occupancy and its effects in Fig S7 doesn't work well. Tons of color and detail but very hard to decipher and intuit what Syt7 is doing to the SV buried in the arrow lengths. And this is a crucial point of the paper - it really needs to shine through in this figure.

We thank the reviewer for pointing out the unclarities in the description of the model. We have worked on improving this section. Specifically, we have improved the equations and now more clearly explain the symbols used in these equations. We have altered the graphical representation of the effect of calcium binding to syt7 on docking and undocking rates.

(8) I would strongly recommend abandoning this large-scale soft modeling effort altogether, but if the authors feel that all the states and parameters are absolutely required, they need to justify this point, define all symbols systematically, number all equations, and provide some evidence of actual data fitting, systematic parameter space exploration, and more exposition of why they are making the various assumptions and constraints that were used to lower the number of free parameters. For instance, why are the tethering and untethering (or docking and undocking) rate constants set to equal each other? And why is it assumed that Syt7 enhances both the docking and undocking rates? Why is fusion set to occur as long as the sum of Syt1 and Doc2 calcium occupancy is exactly 5 regardless of the specific occupancy of either Syt1 or Doc2? Again probably quite important but unjustified physically. Given the efforts of this model to capture some sort of realistic calcium liganding by Syt1, Syt7, and Doc2, the model doesn't seem to take into account the copy number of each protein at a release site. Shouldn't it matter if there are 2 Syt7s vs 20 Syt7s? Or the stoichiometry between Doc2 and Syt1? Either this model assumes that there is exactly one copy of each protein at a release site or that all copies are always identically liganded and strictly act as a unit. Neither of these possibilities seems plausible.

Despite the fact that this model (as all models) is a simplified version of reality and despite the fact that this model (as all models) has its limitations, we decided to keep the model in our work to illustrate that this well-defined hypothesis put forth in this paper is consistent with the experimental data. Again, we are not claiming that this model is the only one that may explain this, nor do we claim that we have uniquely identified its parameters. As indicated above, we worked on improving the description of the model in the methods and improved on our description of how the parameter values are constrained. For the reasons mentioned above (first and foremost because of infeasibility due to excessive computation time) we did not perform data fitting or changed the parameter space exploration. We would like to thank the reviewer for pointing out that some of the assumptions of the model are not well enough explained. We added an extra explanation of these assumptions to the main text.

One of the assumptions we made, as the reviewer points out, is that the tethering and untethering and docking and undocking rates constants are set to equal each other. This is indeed an arbitrary assumption, with the main aim of reducing the number of free parameters in our model given that there is currently no experimental constraint on the relation between the two rate constants. We agree that this assumption is as good as any other, and we have pointed this out more clearly in the main text.

In the model syt7 enhances both docking and undocking rates as we assumed it to function as a catalyst of the docking reaction. A catalyst lowers the energy barrier for the reaction and thereby promotes both forward and backward rates. One of the main reasons we decided on this is because in the model also syt1 and Doc2 are assumed to function by lowering the energy barrier for the fusion reaction. However, since fusion is irreversible this would only affect the forward reaction rate. We cannot exclude that syt7 acts on the forward rate only, which we now mention in the results section of the model.

In our model fusion can occur from any possible docked SV state. The probability of fusion however increases the more calcium ions are bound to Doc2 or Syt1, with Syt1-bound to Calcium being more effective in promoting fusion. This structure matches the dual-sensor model proposed by Sun et al., 2007, Science (PMID: 18046404) and Kobbersmed et al. 2020, Elife (PMID: 32077852), and is based on the assumption that each protein bound to calcium lowers the energy barrier with a certain amount. We have explained this more in the results section of the model.

We decided that syt1 and Doc2 together could have no more than five calcium ions bound to them. This is based on the idea that syt1 and Doc2 are competing for the same type of resources, which could for instance be a limited number of SNARE complexes that are available to execute the reaction. An indication for competition between the two proteins can be found in the synchronous release amplitudes after stimulus 2, which are larger in the Doc2KO.

The reviewer rightfully points out that for realistic simulations of the role of syt1, syt7 and Doc2 the stoichiometry of these proteins at the release site is relevant. In the ideal scenario, we would have included this in our model. However, this would massively increase the possible number of states (which this reviewer criticizes already in our simpler model), making the model even more computationally expensive to run. Additionally, we currently have no reliable estimates of the number of syt7 and Doc2 molecules per release site. In our model, all syt1s expressed on an SV can bind up to five calcium ions. We have recently shown that this simplified model can capture the features of all syt1 proteins per vesicle that compete for the binding of three substrates on the plasma membrane to exert their function in speeding up fusion (Kobbersmed et al., 2022 eLife PMID: 35929728). This means that the copy number is indirectly covered in our model. This number of five calcium ions (and two for Doc2 and syt7) however is not based on the estimated number of syt1s on an SV (which would be around 15, Takamori 2006), but rather on the calcium-dependence of the fusion reaction. Similarly, the number of two calcium ions binding to Doc2 is based on the Calcium-dependence of asynchronous fusion rates (Sun et al., 2007). Based on the reviewer’s comment we now more explicitly mention in the text that the numbers of calcium ions binding to syt1, Doc2 and syt7 corresponds to the total number of calcium ions that can bind to each of these molecules per release site/SV.

We again would like to thank the reviewer for asking us to improve the explanation on the assumptions made to construct our model and how we constrained the parameter values in our model.